# On UMAP's True Loss Function

**Sebastian Damrich**      **Fred A. Hamprecht**
HCI/IWR at Heidelberg University, 69120 Heidelberg, Germany
{sebastian.damrich, fred.hamprecht}@iwr.uni-heidelberg.de

## Abstract

UMAP has supplanted $t$-SNE as state-of-the-art for visualizing high-dimensional datasets in many disciplines, but the reason for its success is not well understood. In this work, we investigate UMAP's sampling based optimization scheme in detail. We derive UMAP's true loss function in closed form and find that it differs from the published one in a dataset size dependent way. As a consequence, we show that UMAP does not aim to reproduce its theoretically motivated high-dimensional UMAP similarities. Instead, it tries to reproduce similarities that only encode the $k$ nearest neighbor graph, thereby challenging the previous understanding of UMAP's effectiveness. Alternatively, we consider the implicit balancing of attraction and repulsion due to the negative sampling to be key to UMAP's success. We corroborate our theoretical findings on toy and single cell RNA sequencing data.

## 1   Introduction

Today's most prominent methods for non-parametric, non-linear dimension reduction are $t$-Distributed Stochastic Neighbor Embedding ($t$-SNE) [20, 19] and Uniform Manifold Approximation and Projection for Dimension Reduction (UMAP) [13]. The heart of UMAP is claimed to be its sophisticated method for extracting the high-dimensional similarities. However, the reason for UMAP's excellent visualizations is not immediately obvious from this approach. In particular, UMAP's eponymous uniformity assumption, see Section 3, is arguably difficult to defend for the wide range of datasets on which UMAP performs well. Therefore, it is not well understood what aspect of UMAP is responsible for its great visualizations.

Both $t$-SNE and UMAP have to overcome the computational obstacle of considering the quadratic number of interactions between all pairs of points. The breakthrough for $t$-SNE came with a Barnes-Hut approximation [19]. Instead, UMAP employs a sampling based approach to avoid a quadratic number of repulsive interactions. Other than [4] little attention has been paid to this sampling based optimization scheme. In this work, we fill this gap and analyze UMAP's optimization method in detail. In particular, we derive the effective, closed form loss function which is truly minimized by UMAP's optimization scheme. While UMAP's use of negative sampling was intended to avoid quadratic complexity, we find, surprisingly, that the resulting effective loss function differs significantly from UMAP's purported loss function. The weight of the loss function's repulsive term is drastically reduced. As a consequence, UMAP is not actually geared towards reproducing the clever high-dimensional similarities. In fact, we show that most information beyond the shared $k$NN graph connectivity is essentially ignored as UMAP actually approximates a binarized version of the high-dimensional similarities. These theoretical findings underpin some empirical observations in [4] and demonstrate that the gist of UMAP is not its high-dimensional similarities. This resolves the disconnect between UMAP's uniformity assumption and its success on datasets of varying density.

From a user's perspective it is important to gain an intuition for deciding which features of a visualization can be attributed to the data and which ones are more likely artifacts of the visualization method. With our analysis, we can explain UMAP's tendency to produce crisp, over-contracted substructures, which increases with the dataset size, as a side effect of its optimization.

35th Conference on Neural Information Processing Systems (NeurIPS 2021).

Without the motivation of reproducing sophisticated high-dimensional similarities in embedding space, it seems unclear why UMAP performs well. We propose an alternative explanation for UMAP's success: The sampling based optimization scheme balances the attractive and repulsive loss terms despite the sparse high-dimensional attraction. Consequently, UMAP can leverage the connectivity information of the shared $k$NN graph via gradient descent effectively.

## 2 Related work

For most of the past decade $t$-SNE [20, 19] was considered state-of-the-art for non-linear dimension reduction. In the last years UMAP [13] at least ties with $t$-SNE. In both methods, points are embedded so as to reproduce high-dimensional similarities. Different from $t$-SNE, the high-dimensional similarities are sparse for UMAP and do not need to be normalized over the entire dataset. Additionally, $t$-SNE adapts the local scale of high-dimensional similarities by achieving a predefined perplexity, while UMAP uses its uniformity assumption. The low-dimensional similarity functions also differ. Recently, Böhm et al. [4] placed both UMAP and $t$-SNE on a spectrum of dimension reduction methods that mainly differ in the amount of repulsion employed. They argue that UMAP uses less repulsion than $t$-SNE. A parametric version of UMAP was proposed in [17].

UMAP's success, in particular in the biological community [2, 16], sparked interest in understanding UMAP more deeply. The original paper [13] motivates the choice of the high-dimensional similarities using concepts from algebraic topology and thus justifies UMAP's transition from local similarities $\mu_{i \to j}$ to global similarities $\mu_{ij}$. The authors find that while the algorithm focuses on reproducing the local similarity pattern similar to $t$-SNE, it achieves better global results. In contrast, Kobak and Linderman [10] attribute the better global properties of UMAP visualizations to the more informative initialization and show that $t$-SNE manages to capture more global structure if initialized in a similar way. Wang et al. [21] also analyze the local and global properties of $t$-SNE, UMAP and TriMAP [1].

Narayan et al. [14] observe that UMAP's uniformity assumption leads to visualizations in which denser regions are more spread out while sparser regions get overly contracted. They propose an additional loss term that aims to reproduce the local density around each point and thus spaces sparser regions out. We provide an additional explanation for overly contracted visualizations: UMAP does not reproduce the high-dimensional similarities but exaggerates the attractive forces over the repulsive ones, which can result in overly crisp visualizations, see Figures 1b, 1c and 2a.

Our work aligns with Böhm et al. [4]. The authors conjecture that the sampling based optimization procedure of UMAP prevents the minimization of the supposed loss function, thus not reproducing the high-dimensional similarities in embedding space. They substantiate this hypothesis by qualitatively estimating the relative size of attractive and repulsive forces. In addition, they implement a Barnes-Hut approximation to the loss function (6) and find that it yields a diverged embedding. We analyze UMAP's sampling procedure in depth, compute UMAP's true loss function in closed form and contrast it against the supposed loss in Section 5. Based on this analytic effective loss function, we can further explain Böhm et al. [4]'s empirical finding that the specific high-dimensional similarities provide little gain over the binary weights of a shared $k$NN graph,[1] see Section 6. Finally, our theoretical framework leads us to a new tentative explanation for UMAP's success in Section 7.

## 3 Background on UMAP

UMAP assumes that the data lies on a low-dimensional Riemannian manifold $(R, g)$ in ambient high-dimensional space. Moreover, the data is assumed to be uniformly distributed with respect to the metric tensor $g$, or put simply, with respect to the distance along the manifold. The metric tensor $g$ in turn is assumed to be locally constant. The key idea of UMAP [13] is to compute pairwise similarities in high-dimensional space which inform the optimization of the low-dimensional embedding. Let $x_1, ..., x_n \in \mathbb{R}^D$ be high-dimensional, mutually distinct data points for which low-dimensional embeddings $e_1, ..., e_n \in \mathbb{R}^d$ shall be found, where $d \ll D$, often $d = 2$ or 3.

First, UMAP extracts high-dimensional similarities between the data points. To do so, the $k$ nearest neighbor graph is computed. Let $i_1, ..., i_k$ denote the indices of $x_i$'s $k$ nearest neighbors in increasing order of distance to $x_i$. Then, using its uniformity assumption, UMAP fits a local notion of similarity

---

[1]The shared $k$ nearest neighbor graph contains an edge $ij$ if $i$ is among $j$'s $k$ nearest neighbors or vice versa.

for each data point $i$ by selecting a scale $\sigma_i$ such that the total similarity of each point to its $k$ nearest neighbors is normalized, i.e. find $\sigma_i$ such that

$$\sum_{\kappa=1}^{k} \exp\left(-\left(d(x_i, x_{i_\kappa}) - d(x_i, x_{i_1})\right)/\sigma_i\right) = \log_2(k). \tag{1}$$

This defines the directed high-dimensional similarities

$$\mu_{i \to j} = \begin{cases} \exp\left(-\left(d(x_i, x_j) - d(x_i, x_{i_1})\right)/\sigma_i\right) \text{ for } j \in \{i_1, \ldots, i_k\} \\ 0 \text{ else.} \end{cases} \tag{2}$$

Finally, these are symmetrized to obtain undirected high-dimensional similarities or input similarities between items $i$ and $j$

$$\mu_{ij} = \mu_{i \to j} + \mu_{j \to i} - \mu_{i \to j}\mu_{j \to i} \in [0, 1]. \tag{3}$$

While each node has exactly $k$ non-zero directed similarities $\mu_{i \to j}$ to other nodes which sum to $\log_2(k)$, this does not hold exactly after symmetrization. Nevertheless, typically the $\mu_{ij}$ are highly sparse, each node has positive similarity to about $k$ other nodes and the degree of each node $d_i = \sum_{j=1}^{n} \mu_{ij}$ is approximately constant and close to $\log_2(k)$, see Supp. Fig. 5 and 6 in Appendix B. For convenience of notation, we set $\mu_{ii} = 0$ and define the total similarity as $\mu_{\text{tot}} = \frac{1}{2}\sum_{i=1}^{n} d_i$.

Distance in embedding space is transformed to low-dimensional similarity by a smooth approximation to the high-dimensional similarity function, $\phi(d; a, b) = (1 + ad^{2b})^{-1}$, for all pairs of points. The shape parameters $a, b$ are essentially hyperparameters of UMAP. We will overload notation and write

$$\nu_{ij} = \phi(||e_i - e_j||) = \phi(e_i, e_j) \tag{4}$$

for the low-dimensional or embedding similarities and usually suppress their dependence on $a$ and $b$.

Supposedly, UMAP approximately optimizes the following objective function with respect to the embeddings $e_1, \ldots, e_n$:

$$\mathcal{L}(\{e_i\}|\{\mu_{ij}\}) = -2 \sum_{1 \le i < j \le n} \left(\mu_{ij}\log(\nu_{ij}) \qquad + (1 - \mu_{ij})\log(1 - \nu_{ij})\right) \tag{5}$$

$$= -2 \sum_{1 \le i < j \le n} \left(\mu_{ij}\underbrace{\log(\phi(e_i, e_j))}_{-\mathcal{L}_{ij}^a} + (1 - \mu_{ij})\underbrace{\log(1 - \phi(e_i, e_j))}_{-\mathcal{L}_{ij}^r}\right). \tag{6}$$

While the high-dimensional similarities $\mu_{ij}$ are symmetric, UMAP's implementation does consider their direction during optimization. For this reason, our losses in equations (5) and (6) differ by a factor of 2 from the one given in [13]. Viewed through the lens of a force-directed model, the derivative of the first term in each summand of the loss function, $-\partial\mathcal{L}_{ij}^a/\partial e_i$, captures the attraction of $e_i$ to $e_j$ due to the high-dimensional similarity $\mu_{ij}$ and the derivative of the second term, $-\partial\mathcal{L}_{ij}^r/\partial e_i$, represents the repulsion that $e_j$ exerts on $e_i$ due to a lack of similarity in high dimension, $1 - \mu_{ij}$. Alternatively, the loss can be seen as the sum of binary cross entropy losses for each pairwise similarity. Thus, it is minimized if the low-dimensional similarities $\nu_{ij}$ exactly match their high-dimensional counterparts $\mu_{ij}$, that is, if UMAP manages to reproduce the input similarities in embedding space.

UMAP uses a sampling based stochastic gradient descent to optimize the low-dimensional embedding typically starting from a Laplacian Eigenmap layout [3, 10]. Our main contribution is to show that the sampling based optimization in fact leads to a different objective function, so that UMAP does not reproduce the high-dimensional similarities in low-dimensional space, see Sections 4 to 6.

## 4    UMAP does not reproduce high-dimensional similarities

UMAP produces scientifically useful visualizations in many domains and is fairly robust to its hyperparameters. Since any visualization of intrinsically high-dimensional data must be somehow unfaithful, it is not straightforward to check the visualization quality other than by its downstream use. Some quantitative metrics such as the correlation between high- and low-dimensional distances [10, 2] exist. We follow a different route to show unexpected properties of UMAP. Consider the toy example of applying UMAP to data that is already low-dimensional, so that no reduction in dimension is

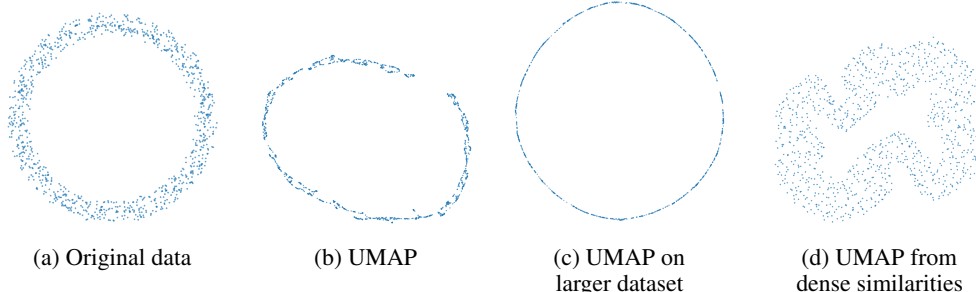

| (a) Original data | (b) UMAP | (c) UMAP on larger dataset | (d) UMAP from dense similarities |

Figure 1: UMAP does not preserve the data even when no dimension reduction is required. **1a:** Original data consisting of 1000 points sampled uniformly from a ring in 2D. All optimizations ran for 10000 epochs and were initialized at the input data. **1b:** Result of UMAP. The circular shape is visible but the ring width is nearly completely contracted. **1c:** Result of UMAP on a dataset consisting of twenty rings as in 1a spaced so they do not interact in the input similarity computation. Only the embedding of one ring is shown. There should not be a significant difference from Fig. 1b. Yet, the ring width is completely contracted. **1d:** Result of UMAP for dense input similarities computed from the original data in Fig. 1a with the similarity function $\phi$ from Section 3. No change from the initialization would be optimal in this setting. Instead the output has spurious curves and larger width. More figures with the default number of epochs and initialization can be found in Supp. Fig. 20.

required: $D = d = 2$. Ideally, the data would be preserved in this situation. One might also expect that UMAP achieves its aim of perfectly reproducing the input similarities. Surprisingly, neither of these expectations is met. In Figure 1, we depict 2D UMAP visualizations of a 2D uniform ring dataset. To ease UMAP's task, we initialized the embedding with the original data, hoping that UMAP's optimization would just deem this layout to be optimal. We also used a longer run time to ensure convergence of UMAP's optimization procedure. Otherwise, we used the default hyperparameters. The results with default initialization and run time are qualitatively similar and shown in Supp. Fig. 20. UMAP manages to capture the ring shape of the data, but changes its appearance significantly. It contracts the width of the ring nearly to a line, see Figure 1b, a phenomenon also observed on real-world datasets, see Figure 2a. Whether this exaggeration of the ring shape is useful depends on the use case. This finding goes beyond Narayan et al. [14]'s observation that over-contraction happens in regions of low density since our toy dataset is sampled uniformly from a circular ring.

Moreover, we embedded the same ring with UMAP as part of a larger dataset of twenty such rings, spaced so that the input similarities for each ring are not influenced by the other rings. According to Section 3, we would expect a qualitatively similar embedding as for the individual ring. However, we see in Figure 1c that UMAP produces an even crisper embedding. In another experiment, we varied the number of samples from a single ring and observed again that the tendency for over-contraction increases with the dataset size. For more information on UMAP's dependence on the dataset size confer Appendix H. The dataset size dependence is particularly noteworthy, as computing UMAP embeddings of subsets or subsamples of a dataset is common practice.

As described in Section 3, UMAP employs different methods in input and embedding space to transform distances to similarities. In particular, in input space similarities are zero for all but the closest neighbors, while in embedding space they are computed with the heavy-tailed function $\phi$. To test whether this prevents the reproduction of the input data, we also computed the dense similarities on the original data with $\phi$ and used these as input similarities for the embedding in Figure 1d. Since we also initialize with the original dataset, a global optimum of the objective function (6) for this choice of input similarity, one would expect no change by UMAP's optimization scheme. However, we observe that in this setting UMAP produces spurious curves and increases the width of the ring.

Böhm et al. [4] implemented a Barnes-Hut approximation of UMAP's objective function (6), which produced a diverged embedding. Inspired by this finding, we compute the loss values according to equation (5) for various input and embedding similarities $\mu_{ij}$ and $\nu_{ij}$ in our toy example, see Table 1.

Consider the row with the usual input similarities ($\mu_{ij} = \mu(\{x_1, \ldots, x_n\})$). In a completely diverged embedding, all self similarities are one and all others zero ($\nu_{ij} = \mathbb{1}(i == j)$). We find that the loss for such an embedding is lower than for the optimized UMAP embedding. This is in

Table 1: UMAP loss value for various combinations of input and embedding similarities, $\mu_{ij}, \nu_{ij}$, of the toy example in Figure 1. The loss for the UMAP embedding in Figure 1b (middle column) is always higher than for another two-dimensional layout (**bold**). Hence, UMAP does not minimize its purported loss. Results are averaged over 7 runs and one standard deviation is given, see also Appendix F.1.

| | Embedding similarities $\nu_{ij}$ | | |
| Input similarities $\mu_{ij}$ | $\mathbb{1}(i == j)$ (diverged layout) | $\phi(\{e_1, \ldots, e_n\})$ (UMAP result) | $\phi(\{x_1, \ldots, x_n\})$ (input layout) |
|---|---|---|---|
| $\mu(\{x_1, \ldots, x_n\})$ | $\mathbf{62959 \pm 82}$ | $70235 \pm 1301$ | $136329 \pm 721$ |
| $\phi(\{x_1, \ldots, x_n\})$ | $902757 \pm 2788$ | $331666 \pm 1308$ | $\mathbf{224584 \pm 8104}$ |

accordance with Böhm et al. [4]'s Barnes-Hut experiment and shows that UMAP does not optimize its supposed objective function (6) as a diverged embedding is approximately feasible in two dimensions. This discrepancy is not just due to the fact that input and embedding similarities are computed differently: The second row of Table 1 contains loss values for the setting in which we use the dense similarities as input similarities, as in Figure 1d. We initialize the embedding at the optimal loss value ($\nu_{ij} = \phi(\{x_1, \ldots, x_n\}) = \mu_{ij}$), but UMAP's optimization moves away from this layout and towards an embedding with higher loss ($\nu_{ij} = \phi(\{e_1, \ldots, e_n\})$) although we always compute similarity in the same manner. Clearly, UMAP's optimization yields unexpected results.

## 5 UMAP's sampling strategy and effective loss function

UMAP uses a sampling based approach to optimize its loss function, to reduce complexity. A simplified version of the sampling procedure can be found in Algorithm 1. Briefly put, an edge $ij$ is sampled according to its high-dimensional similarity and the embeddings $e_i$ and $e_j$ of the incident nodes are pulled towards each other. For each such sampled edge $ij$, the algorithm next samples $m$ negative samples $s$ uniformly from all nodes and the embedding of $i$ is repelled from that of each negative sample. Note that the embeddings of the negative samples are not repelled from that of $i$, see Alg. 1 line 10. So there are three types of gradient applied to an embedding $e_i$ during an epoch:

1. $e_i$ is pulled towards $e_j$ when edge $ij$ is sampled (Alg.1 line 5)
2. $e_i$ is pulled towards $e_j$ when edge $ji$ is sampled (Alg.1 line 6)
3. $e_i$ is pushed away from negative sample embeddings when some edge $ij$ is sampled (Alg.1 line 9).

The full update of embedding $e_i$ during epoch $t$ is, according to UMAP's implementation, given by

$$g_i^t = \sum_{j=1}^{n} \left( X_{ij}^t \cdot \frac{\partial \mathcal{L}_{ij}^a}{\partial e_i} + X_{ji}^t \cdot \frac{\partial \mathcal{L}_{ji}^a}{\partial e_i} + X_{ij}^t \cdot \sum_{s=1}^{n} Y_{ij,s}^t \cdot \frac{\partial \mathcal{L}_{is}^r}{\partial e_i} \right), \tag{7}$$

where $X_{ab}^t$ is the binary random variable indicating whether edge $ab$ was sampled in epoch $t$ and $Y_{ab,s}^t$ is the random variable for the number of times $s$ was sampled as negative sample for edge $ab$ in epoch $t$ if $ab$ was sampled in epoch $t$ and zero otherwise. By construction, $\mathbb{E}(X_{ab}^t) = \mu_{ab}$ and $\mathbb{E}(Y_{ab,s}^t | X_{ab}^t = 1) = m/n$. Taking the expectation over the random events in an epoch, we obtain the expected update of UMAP's optimization procedure, see Appendix C for more details:

$$\mathbb{E}\left(g_i^t\right) = 2 \sum_{j=1}^{n} \left( \mu_{ij} \cdot \frac{\partial \mathcal{L}_{ij}^a}{\partial e_i} + \frac{d_i m}{2n} \cdot \frac{\partial \mathcal{L}_{ij}^r}{\partial e_i} \right). \tag{8}$$

Comparing the above closed formula for the expectation of the UMAP updates of the low-dimensional embeddings, to the gradient of UMAP's loss function (6)

$$\frac{\partial \mathcal{L}}{\partial e_i} = 2 \sum_{j=1}^{n} \left( \mu_{ij} \cdot \frac{\partial \mathcal{L}_{ij}^a}{\partial e_i} + (1 - \mu_{ij}) \cdot \frac{\partial \mathcal{L}_{ij}^r}{\partial e_i} \right), \tag{9}$$

we find that the sampling procedure yields the correct weight for the attractive term in expectation, as designed. However, as noticed by Böhm et al. [4], the negative sampling changes the weight for the

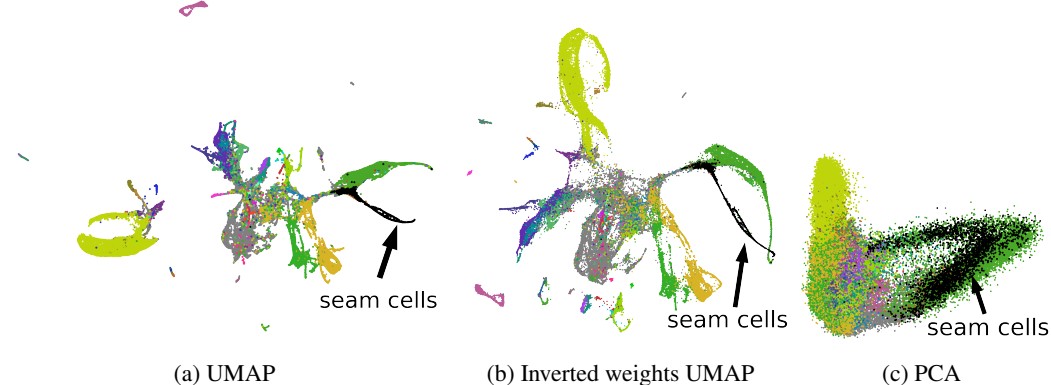

|  |  |  |
|:---:|:---:|:---:|
| (a) UMAP | (b) Inverted weights UMAP | (c) PCA |

Figure 2: UMAP on C. elegans data from [16, 14]. **2a:** UMAP visualization. Several parts of the embedding appear locally one-dimensional, for instance the seam cells.[2]
**2b:** Same as 2a but with inverted positive high-dimensional similarities. The result is qualitatively similar, if not better. **2c:** Two dimensional PCA of the dataset. Highlighted seam cells clearly have two dimensional variance in the PCA plot, but are overly contracted to nearly a line in the UMAP plots 2a and 2b. The full legend with all cell types and further information can be found in Supp. Fig. 14. We report quantitative metrics in Appendix G. Figure best viewed in color.

repulsive term significantly. Our closed formula helps to make their qualitative arguments precise: Instead of $1 - \mu_{ij}$, we have a term $\frac{d_i m}{2n}$, which depends on the number of negative samples $m$ per sampled edge. Contrary to the intention of McInnes et al. [13], the repulsive weights are not uniform but vary with the degree of each point $d_i$, which, however, is typically close to $\log_2(k)$, see Appendix B. More practically, since the non-zero high-dimensional similarities are sparse, $1 - \mu_{ij}$ is equal to 1 for most $ij$. In contrast, the expected repulsive weight is small for large datasets as the numerator $d_i m$ is of the order of $\log_2(k)m$ independent of the dataset size $n$ but the denominator scales with $n$.

Another effect of the negative sampling is that in general the expected update (8) does not correspond to any loss function, see Appendix D. We remedy this by additionally pushing the embedding of a negative sample $i$ away from the embedding $e_j$, whenever $i$ was sampled as negative sample to some edge $jk$, see Alg.1 line 10. This yields the following gradient at epoch $t$

$$\tilde{g}_i^t = \sum_{j=1}^n \left( X_{ij}^t \cdot \frac{\partial \mathcal{L}_{ij}^a}{\partial e_i} + X_{ji}^t \cdot \frac{\partial \mathcal{L}_{ji}^a}{\partial e_i} + X_{ij}^t \cdot \sum_{s=1}^n Y_{ij,s}^t \cdot \frac{\partial \mathcal{L}_{is}^r}{\partial e_i} + \sum_{k=1}^n X_{jk}^t Y_{jk,i}^t \cdot \frac{\partial \mathcal{L}_{ji}^r}{\partial e_i} \right), \quad (10)$$

corresponding to a loss in epoch $t$ of

$$\tilde{\mathcal{L}}^t = \sum_{1 \le i,j \le n} \left( X_{ij}^t \cdot \mathcal{L}_{ij}^a + \sum_{s=1}^n X_{ij}^t Y_{ij,s}^t \cdot \mathcal{L}_{is}^r \right). \quad (11)$$

Using the symmetry of $\mu_{ij}$, $\mathcal{L}_{ij}^a$ and $\mathcal{L}_{ij}^r$ in $i$ and $j$, we compute the effective loss

$$\tilde{\mathcal{L}} = \mathbb{E}(\tilde{\mathcal{L}}^t) = 2 \sum_{1 \le i < j \le n} \left( \mu_{ij} \cdot \mathcal{L}_{ij}^a + \frac{(d_i + d_j)m}{2n} \cdot \mathcal{L}_{ij}^r \right). \quad (12)$$

In fact, the above remedy of also pushing the negative samples does not affect the behavior of UMAP qualitatively, see for instance Supp. Figs. 21 and 15.[3] In this light, we can treat $\tilde{\mathcal{L}}$ as the effective objective function that is optimized via SGD by UMAP's optimization procedure. It differs from UMAP's loss function (6) by having a drastically reduced repulsive weight of $\frac{(d_i + d_j)m}{2n}$ instead of $1 - \mu_{ij}$.

We illustrate our analysis on gene expression measurements of 86024 cells of C. elegans [16, 14]. We start out with a 100 dimensional PCA of the data[4] and use the cosine metric in high-dimensional space,

---

[2]Near the tip of the seam cells, the points seem to lie on a one-dimensional manifold. Closer to the middle of the seam cells the spread appears wider. But there are actually two dense parallel lines of points.

[3]In fact, the Parametric UMAP [17] does include the update of negative samples.

[4] obtained from `http://cb.csail.mit.edu/cb/densvis/datasets/`. We informed the authors of our use of the dataset, which they license under CC BY-NC 2.0.

consider $k = 30$ neighbors and optimize for 750 epochs, similar to [14]. The resulting visualization is depicted in Figure 2a. On this dataset the average value of $1 - \mu_{ij}$ is 0.9999 but the maximal effective repulsive weight $\max_{ij} \frac{(d_i + d_j)m}{2n}$ is 0.0043, showing the dramatic reduction of repulsion due to negative sampling. After each optimization epoch, we log our effective loss $\tilde{\mathcal{L}}$ (12), the actual loss $\tilde{\mathcal{L}}^t$ (11) of each epoch computed based on the sampled (negative) pairs as well the purported UMAP loss $\mathcal{L}$ (6).[5] We always consider the embeddings at the end of each epoch. Note that UMAP's implementation updates each embedding $e_i$ not just once at the end of the epoch but as soon as $i$ is incident to a sampled edge or sampled as a negative sample. This difference does not change the actual loss much, see Appendix F. The loss curves are plotted in Figure 3. We can see that our predicted loss matches its actual counterpart nearly perfectly. While both, $\tilde{\mathcal{L}}$ and $\tilde{\mathcal{L}}^t$, agree with the attractive part of the supposed UMAP loss, its repulsive part and thus the total loss are two orders of magnitude larger. Furthermore, driven by the repulsive part, the total intended UMAP loss increases during much of the optimization process, while the actual and effective losses decrease, exemplifying that UMAP really optimizes our effective loss $\tilde{\mathcal{L}}$ (12) instead of its purported loss $\mathcal{L}$ (6). Additional loss curves for UMAP on other scRNA-seq datasets and CIFAR-10 [11] can be found in Appendix I.

Recently, a parametric version of UMAP was proposed [17]. Here, a neural network is trained to perform the mapping from high to low dimension. Its effective loss function differs slightly from the one for Non-Parametric UMAP as negative samples only come for the training batch:

**Theorem 5.1.** *The effective loss function of Parametric UMAP with neural network $f_\theta$, batch size $b$ and total similarity $\mu_{\mathrm{tot}} = \frac{1}{2} \sum_{i=1}^n d_i$ is*

$$-\frac{1}{(m+1)\mu_{\mathrm{tot}}} \sum_{1 \le i < j \le n} \mu_{ij} \log\Big(\phi\big(f_\theta(x_i), f_\theta(x_j)\big)\Big) + m \frac{b-1}{b} \frac{d_i d_j}{2\mu_{\mathrm{tot}}} \log\Big(1 - \phi\big(f_\theta(x_i), f_\theta(x_j)\big)\Big).$$

$$(13)$$

*Proof.* The proof can be found in Appendix A. □

While Parametric UMAP's loss differs slightly from $\tilde{\mathcal{L}}$ (12) the same analysis holds unless explicitly mentioned. We mention a relation of Parametric UMAP to modularity clustering in Appendix A.1.

---

**Algorithm 1:** UMAP's optimization

**input** : input similarities $\mu_{ij}$,
    initial embeddings $e_1, \dots, e_n$,
    number of epochs T,
    learning rate $\alpha$,
    number of negative samples $m$
**output** : final embeddings $e_1, \dots, e_n$

1 **for** $t = 0$ **to** $T$ **do**
2  **for** $ij \in \{1, \dots, n\}^2$ **do**
3   $r \sim \text{Uniform}(0, 1)$
4   **if** $r < \mu_{ij}$ **then**
5    $e_i = e_i - \alpha \cdot \frac{\partial \mathcal{L}_{ij}^a}{\partial e_i}$
6    $e_j = e_j - \alpha \cdot \frac{\partial \mathcal{L}_{ij}^a}{\partial e_j}$
7    **for** $l = 1$ **to** $m$ **do**
8     $s \sim \text{Uniform}(\{1, \dots, n\})$
9     $e_i = e_i - \alpha \cdot \frac{\partial \mathcal{L}_{is}^r}{\partial e_i}$
     `// Next line is omitted`
     `in UMAP's implementation,`
     `but included for our`
     `analysis`
10     /* $e_s = e_s - \alpha \cdot \frac{\partial \mathcal{L}_{is}^r}{\partial e_s}$ */

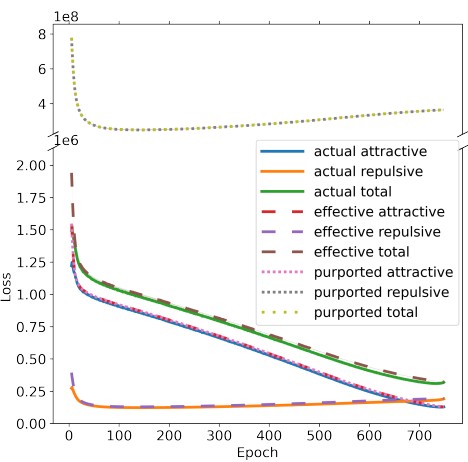

Figure 3: Loss curves for the optimization leading to Figure 2a. Our effective loss (12) closely matches the actual loss (11), while the purported UMAP loss (6) is two orders of magnitude higher. The total overlays the repulsive purported loss. An average over 7 runs is plotted with shaded areas of one standard deviation, see also Appendix F.1. Figure best viewed in color.

---

[5]Our code is publicly available at `https://github.com/hci-unihd/UMAPs-true-loss`.

# 6 True target similarities

Since the effective objective function $\tilde{\mathcal{L}}$ (12) that UMAP optimizes is different from $\mathcal{L}$ (6), we cannot hope that UMAP truly tries to find a low-dimensional embedding whose similarities reproduce the high-dimensional similarities. Nevertheless, using the effective loss $\tilde{\mathcal{L}}$, we can compute the true target similarities $\nu_{ij}^*$ which UMAP tries to achieve in embedding space. The effective loss $\tilde{\mathcal{L}}$ is a sum of non-normalized binary cross entropy loss functions

$$-\left(\mu_{ij} \cdot \log(\nu_{ij}) + \frac{(d_i + d_j)m}{2n} \cdot \log(1 - \nu_{ij})\right) \tag{14}$$

which are minimal for

$$\nu_{ij}^* = \frac{\mu_{ij}}{\mu_{ij} + \frac{(d_i + d_j)m}{2n}} \begin{cases} = 0 \text{ if } \mu_{ij} = 0 \\ \approx 1 \text{ if } \mu_{ij} > 0. \end{cases} \tag{15}$$

The approximation holds in the typical case in which $\frac{(d_i+d_j)m}{2n} \approx 0$, discussed above. In other words, compared to the massively reduced repulsion weight even the smallest high-dimensional similarity appears maximal. Thus, the negative sampling essentially binarizes the input similarities. UMAP's high-dimensional similarities are non-zero exactly on the shared $k$ nearest neighbor graph edges of the high-dimensional data. Therefore, the binarization explains why Böhm et al. [4] find that using the binary weights of the shared $k$ nearest neighbor graph does not deteriorate UMAP's performance much.[6] The binarization even helps UMAP to overcome disrupted high-dimensional similarities, as long as only the edges of the shared $k$NN graph have non-zero weight. In Figure 2b, we invert the original positive high-dimensional weights on the C. elegans dataset. That means that the $k$-th nearest neighbor will have higher weight than the nearest neighbor. The resulting visualization even improves on the original by keeping the layout more compact. This underpins Böhm et al. [4]'s claim that the elaborate theory used to compute the high-dimensional similarities is not the reason for UMAP's practical success. In fact, we show that UMAP's optimization scheme even actively ignores most information beyond the shared $k$NN graph. In Figure 4, we show histograms of the various notions of similarity for the C. elegans dataset. We see in panel 4b how the binarization equalizes the positive target similarities for the original and the inverted high-dimensional similarities.

UMAP's tendency for over-contraction is already strong for small datasets. Consider UMAP's default setting of $k = 15$ and $m = 5$. Then for datasets with $n = 500$ points each input similarity $\mu_{ij} > 0.2$ is mapped to a target similarity $\nu_{ij}^* > 0.83$, see Appendix H.4.

The binary cross entropy terms in the effective loss $\tilde{\mathcal{L}}$ (12) are not normalized. This leads to a different weighing of the binary cross entropy terms for each pair $ij$

$$\tilde{\mathcal{L}} = 2 \sum_{1 \le i < j \le n} \left( \mu_{ij} \cdot \mathcal{L}_{ij}^a + \frac{(d_i + d_j)m}{2n} \cdot \mathcal{L}_{ij}^r \right) \tag{16}$$

$$= -2 \sum_{1 \le i < j \le n} \left( \left( \mu_{ij} + \frac{(d_i + d_j)m}{2n} \right) \cdot \left( \nu_{ij}^* \log(\nu_{ij}) + (1 - \nu_{ij}^*) \log(1 - \nu_{ij}) \right) \right). \tag{17}$$

As $\frac{(d_i+d_j)m}{2n}$ is very small for large datasets, the term $\mu_{ij} + \frac{(d_i+d_j)m}{2n}$ is dominated by $\mu_{ij}$. Hence, the reduced repulsion not only binarizes the high-dimensional similarities, it also puts higher weight on the positive than the zero target similarities. Therefore, we can expect that the positive target similarities are better approximated by the embedding similarities than the zero ones. Indeed, panel 4a shows that the low-dimensional similarities match the positive target similarities very well, as expected from the weighted BCE reading of the effective loss function (17).

## 6.1 Explaining artifacts in UMAP visualizations

We conclude this section by explaining the observed artifacts of UMAP's visualization in Figures 1 and 2 in the light of the above analysis. The normal UMAP optimization contracts the ring in Figure 1b even when initialized at the original layout because the reduced repulsion yields nearly binary target similarities. All pairs that are part of the $k$NN graph not only want to be sufficiently

---

[6]Böhm et al. [4] used a scaled version of the $k$NN graph, but the scaling factor cancels for the target weights.

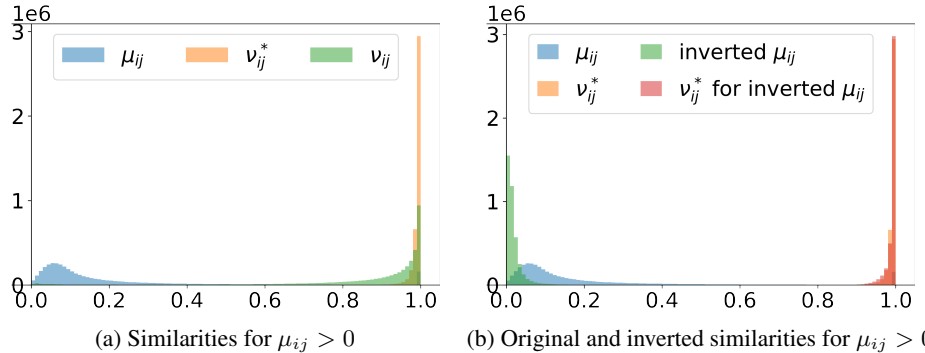

(a) Similarities for $\mu_{ij} > 0$  (b) Original and inverted similarities for $\mu_{ij} > 0$

Figure 4: Histograms of high-dimensional ($\mu_{ij}$), target ($\nu_{ij}^*$) and low-dimensional ($\nu_{ij}$) similarities on the C. elegans dataset [16, 14] for pairs with positive high-dimensional similarity. **4a:** The similarities of UMAP's low-dimensional embedding reproduce the target similarities instead of the high-dimensional ones. The target similarities are heavily skewed towards one. **4b:** Comparison of positive input and target similarities for the original and inverted input similarities. While the histograms of the input similarities differ noticeably, their target similarities do not and neither do the embedding similarities (not shown). The binarization essentially ignores all information beyond the $k$NN graph.

close that their high-dimensional similarity is reproduced, but so close that their similarity is nearly one. The fact that the effective loss weighs the terms with target similarity near one much more than those with target similarity near zero reinforces this trend. As a result, the ring gets contracted. The same argument applies to the over-contracted parts of the UMAP visualization of the C. elegans dataset in Figure 2. When we increase the dataset size by adding more rings as in Figure 1c, negative sample pairs can come from different rings, reducing the frequency that a pair of points from the same ring is sampled. This reduces the repulsion further. Thus, there is stronger binarization and the embedding looks crisper. If we sample a single ring more densely, over-contraction is also increased. But the visual appearance is more nuanced as over-contraction can appear in the form of densely clustered and line-like substructures within the ring width, see Appendix H.3.

Our framework can also explain the opposite behavior of UMAP when the dense similarities are used as input similarities in Figure 1d. In this setting, the average degree of a node is about 100. With a negative sample rate $m = 5$ and a dataset size $n = 1000$ this yields repulsive weights $\frac{(d_i + d_j)m}{2n} \approx 0.5$. Thus, we increase the repulsion on pairs with high input similarity, but decrease it on pairs with low input similarity. Now, the target similarities are not a binarization of the input similarities, but instead skewed towards $0.5$. Hence, we can expect embedding points to increase their distance to nearest neighbors, but distant points to move closer towards each other. This is what we observe in Figure 1d, where the width of the ring has increased and the ring curves to bring distant points closer together.

## 7  Discussion

### 7.1  Negative sampling in LargeVis

The visualization method LargeVis [18] uses negative sampling to optimize its objective function, too. Thus, our analysis of the loss function and the target similarities applies to LargeVis and helps to explain why UMAP and LargeVis visualizations are often similar. For more details confer Appendix E.

### 7.2  Balancing attraction and repulsion

By deriving UMAP's true loss function and target similarities, we are able to explain several peculiar properties of UMAP visualizations. According to our analysis, UMAP does not aim to reproduce the high-dimensional UMAP similarities in low dimension but rather the binary weights of the shared $k$NN graph of the input data. This raises the question just what part of UMAP's optimization leads to its excellent visualization results. Apparently, the exact formula for the repulsive weights is not crucial as it differs for Non-Parametric UMAP and Parametric UMAP while both produce similarly high quality embeddings. A first tentative step towards an explanation might be the different weighing

of the BCE terms in the effective loss function (17). Focusing more on the similar rather than the dissimilar pairs might help to overcome the imbalance between an essentially linear number of attractive and a quadratic number of repulsive pairs. Inflated attraction was found beneficial for $t$-SNE as well, in the form of early exaggeration [12].

Put another way, the decreased repulsive weights result in comparable total attractive and repulsive weights, which might facilitate the SGD based optimization. Indeed, the total attractive weight in UMAP's effective loss function is $2\mu_{\text{tot}} = \sum_{i,j=1}^{n} \mu_{ij}$ and the total repulsive weight amounts to $2m\mu_{\text{tot}} = \sum_{i,j=1}^{n} \frac{(d_i+d_j)m}{2n}$ for Non-Parametric UMAP. For Parametric UMAP the total attractive and repulsive weights are $1/(m+1)$ and $m/(m+1)\cdot(b-1)/b$, respectively. For the default value of $m = 5$, the total attractive and repulsive weights are of roughly the same order of magnitude. Moreover, we observe in Figure 3 that the resulting attractive and repulsive losses are also of comparable size. Using UMAP's purported loss function, however, would yield dominating repulsion.

### 7.3 Improved UMAP

The reason for the binarization of the input similarities is the reduced repulsion on the pairs of points that are linked in the shared $k$NN graph. As discussed above, it might not be desirable to increase the repulsion on all pairs of points to $1 - \mu_{ij}$. But it might be useful to change the optimization procedure of UMAP so as to decrease the repulsion per edge only on the non-$k$NN graph edges and not on the $k$NN graph edges. In addition to the positive and negative sampling of UMAP, we suggest to explicitly sample the shared $k$NN graph edges for repulsion with probability $1 - \mu_{ij} - (d_i + d_j)m/(2n)$. This would correct the repulsion weight for edges in the shared $k$NN graph. The loss function would read

$$-2 \sum_{ij \in k\text{NN graph}} \Big( \mu_{ij} \log(\nu_{ij}) + (1-\mu_{ij}) \log(1-\nu_{ij}) \Big) - 2 \sum_{ij \notin k\text{NN graph}} \left( \frac{(d_i + d_j)m}{2n} \log(1 - \nu_{ij}) \right). \quad (18)$$

Now the target similarities are precisely the high-dimensional similarities $\mu_{ij}$ for all pairs $i, j$. Hence, in an optimal embedding, all $\mu_{ij}$'s are exactly reproduced and not just a binary version of them. We hope that this will improve the embedding quality and decrease the observation of over-contraction. Crucially, the amount of repulsion per data point is kept constant and is not increasing with $n$. For the quadratic number of non-$k$NN graph edges, the individual repulsion pre-factors still decrease with $1/n$. The total loss function is a sum of normalized cross-entropy terms for the edges in the $k$NN graph and of non-normalized and drastically down-weighted cross-entropy terms for the edges not in the $k$NN graph. Since not all $O(n^2)$ negative pairs are considered explicitly and the additional sampling of the $k$NN graph edges for repulsion only scales with $O(n)$, the complexity of computing the embedding would still scale linearly as $O(kmn \cdot \text{num}_{\text{epochs}})$ and not quadratically in $n$.

A more in-depth investigation as to why exactly balanced attraction and repulsion is beneficial for a useful embedding and the implementation of the improved sampling scheme are left for future work.

## 8 Conclusion

In this work, we investigated UMAP's optimization procedure in depth. In particular, we computed UMAP's effective loss function analytically and found that it differs slightly between the non-parametric and parametric versions of UMAP and significantly from UMAP's alleged loss function. The optimal solution of the effective loss function is typically a binarized version of the high-dimensional similarities. This shows why the sophisticated form of the high-dimensional UMAP similarities does not add much benefit over the shared $k$NN graph. Instead, we conjecture that the resulting balance between attraction and repulsion is the main reason for UMAP's great visualization capability. Our analysis can explain some artifacts of UMAP visualizations, in particular its tendency to produce over-contracted embeddings which gets stronger for larger datasets.

## Acknowledgments and Disclosure of Funding

We would like to thank Dmitry Kobak, Quentin Garrido and the anonymous reviewers for useful discussions and helpful feedback. Moreover, we thank Ashwin Narayan for making the scRNA-seq datasets available.

This work is supported in part by the Deutsche Forschungsgemeinschft (DFG, German Research Foundation) – Projektnummer 240245660 - SFB 1129, the Deutsche Forschungsgemeinschaft (DFG, German Research Foundation) under Germany's Excellence Strategy EXC 2181/1 - 390900948 (the Heidelberg STRUCTURES Excellence Cluster) and Informatics for Life funded by the Klaus Tschira Foundation.

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
