# A Parametric UMAP's sampling and effective loss function

In Parametric UMAP [17] the embeddings are not directly optimized. Instead a parametric function, a neural network, is trained to map the input points to embedding space. As usual, a mini-batch of data points is fed through the neural network at each training iteration. The loss is computed for this mini-batch and then the parameters of the neural network are updated via stochastic gradient descent. To avoid the quadratic complexity of the repulsive term a sampling strategy is employed, sketched in Algorithm 2. There are three differences to the optimization scheme of Non-Parametric UMAP: First, since automatic differentiation is used, not only the head of a negative sample edge is repelled from the tail but both repel each other. Second, the same number of edges are sampled in each epoch. Third, since only the embeddings of the current mini-batch are available, negative samples are produced not from the full dataset but only from within the non-uniformly assembled batch. This leads to a different repulsive weight for Parametric UMAP as described in

**Theorem A.1.** *The expected loss function of Parametric UMAP is*

$$-\frac{1}{2(m+1)\mu_{\text{tot}}} \sum_{i,j=1}^{n} \Bigg( \mu_{ij} \cdot \log\Big( \phi\big(f_\theta(x_i), f_\theta(x_j)\big) \Big)$$

$$+ m\frac{b-1}{b} \frac{d_i d_j}{2\mu_{\text{tot}}} \cdot \log\Big(1 - \phi\big(f_\theta(x_i), f_\theta(x_j)\big)\Big) \Bigg), \tag{19}$$

*where $m$ is the negative sample rate, $\mu_{\text{tot}} = \frac{1}{2}\sum_{i=1}^{n} d_i$ the total similarity, $b$ the batch size and $f_\theta$ the parametric embedding function.*

*Proof.* Let $P_{ij}$ be the random variable for the number of times that edge $ij$ is sampled into the batch $B$ of some iteration $t$. Let further $N_{ij}$ be the random variable holding the number of negative sample pairs $ij$ in that iteration. Then the loss at iteration $t$ is given by

$$\mathcal{L}^t = -\frac{1}{(m+1)b} \sum_{i,j=1}^{n} \Bigg( P_{ij} \cdot \log\Big( \phi\big(f_\theta(x_i), f_\theta(x_j)\big) \Big) + N_{ij} \cdot \log\Big(1 - \phi\big(f_\theta(x_i), f_\theta(x_j)\big)\Big) \Bigg) \tag{20}$$

To compute the expectation of this loss, we need to find the expectations of the $P_{ij}$'s and $N_{ij}$'s. The edges in batch $B$ are sampled independently with replacement from the categorical distribution over all pairs $rs$ with probability proportional to the high-dimensional similarities. Thus, $P_{ij}$ follows the multinomial distribution $\text{Mult}(b, \{\frac{\mu_{rs}}{2\mu_{\text{tot}}}\}_{r,s=1,\dots,n})$ and $\mathbb{E}(P_{ij}) = \frac{b\mu_{ij}}{2\mu_{\text{tot}}}$.

To get the negative sample pairs, each entry of the heads $B_h$ and tails $B_t$ in $B$ is repeated $m$ times. We introduce the random variables $H_r$ and $T_r$ for $r = 1, \dots, n$, representing the number of occurrences of node $r$ among the repeated heads and tails. $N_{ij}$ counts how often the sampled permutation, $\pi$, of the repeated tails assigns a tail $j$ to a head $i$. This can be viewed as selecting a tail from $mB_t$ (tails repeated $m$ times) for each of the $H_i$ heads $i$ without replacement. There are $T_j$ tails that could lead to a negative sample pair $ij$. Therefore, $N_{ij}$ follows a hypergeometric distribution $\text{Hyp}(mb, H_i, T_j)$. So, $\mathbb{E}_\pi(N_{ij}) = \frac{H_i T_j}{mb}$. We have

$$H_i = m \cdot \sum_{s=1}^{n} P_{is} \text{ and } T_j = m \cdot \sum_{r=1}^{n} P_{rj}. \tag{21}$$

Since the multinomially distributed $P_{rs}$'s have covariance $\text{Cov}(P_{rs}, P_{r's'}) = -b\frac{\mu_{rs}\mu_{r's'}}{4\mu_{\text{tot}}^2}$, we get

$$\mathbb{E}_B(P_{rs}P_{r's'}) = \text{Cov}(P_{rs}, P_{r's'}) + \mathbb{E}_B(P_{rs})\mathbb{E}_B(P_{r's'}) = b(b-1)\frac{\mu_{rs}\mu_{r's'}}{4\mu_{\text{tot}}^2}. \tag{22}$$

With this we compute the expectation of $\mathbb{E}_\pi(N_{ij})$ with respect to the batch assembly as

$$
\begin{aligned}
\mathbb{E}_B(\mathbb{E}_\pi(N_{ij})) &= \frac{1}{mb}\mathbb{E}_B(H_i T_j) \\
&= \frac{1}{mb}\mathbb{E}_B\left(m\sum_{s=1}^n P_{is}\cdot m\sum_{r=1}^n P_{rj}\right) \\
&= \frac{m}{b}\sum_{r,s=1}^n \mathbb{E}_B(P_{is}P_{rj}) \\
&= \frac{m}{b}\sum_{r,s=1}^n b(b-1)\frac{\mu_{is}\mu_{rj}}{4\mu_{\text{tot}}^2} \\
&= m(b-1)\frac{d_i d_j}{4\mu_{\text{tot}}^2}.
\end{aligned}
\tag{23}
$$

Finally, as the random process of the batch assembly is independent of the choice of the permutation, we can split the total expectation up and get the expected loss

$$
\begin{aligned}
&\mathbb{E}_{(B,\pi)}(\mathcal{L}^t) \\
&= \mathbb{E}_B\mathbb{E}_\pi\left(-\frac{1}{(m+1)b}\sum_{i,j=1}^n\left(P_{ij}\cdot\log\left(\phi\big(f_\theta(x_i),f_\theta(x_j)\big)\right)\right.\right. \\
&\hspace{5cm}\left.\left. + N_{ij}\cdot\log\left(1-\phi\big(f_\theta(x_i),f_\theta(x_j)\big)\right)\right)\right) \\
&= -\frac{1}{(m+1)b}\sum_{i,j=1}^n\left(\mathbb{E}_B\mathbb{E}_\pi(P_{ij})\cdot\log\left(\phi\big(f_\theta(x_i),f_\theta(x_j)\big)\right)\right. \\
&\hspace{4cm}\left. + \mathbb{E}_B\mathbb{E}_\pi(N_{ij})\cdot\log\left(1-\phi\big(f_\theta(x_i),f_\theta(x_j)\big)\right)\right) \\
&= -\frac{1}{(m+1)b}\sum_{i,j=1}^n\left(\frac{b\mu_{ij}}{2\mu_{\text{tot}}}\cdot\log\left(\phi\big(f_\theta(x_i),f_\theta(x_j)\big)\right)\right. \\
&\hspace{3.5cm}\left. + m(b-1)\frac{d_i d_j}{4\mu_{\text{tot}}^2}\cdot\log\left(1-\phi\big(f_\theta(x_i),f_\theta(x_j)\big)\right)\right) \\
&= -\frac{1}{2(m+1)\mu_{\text{tot}}}\sum_{i,j=1}^n\left(\mu_{ij}\cdot\log\left(\phi\big(f_\theta(x_i),f_\theta(x_j)\big)\right)\right. \\
&\hspace{3.5cm}\left. + m\frac{b-1}{b}\frac{d_i d_j}{2\mu_{\text{tot}}}\cdot\log\left(1-\phi\big(f_\theta(x_i),f_\theta(x_j)\big)\right)\right).
\end{aligned}
\tag{24}
$$

$\square$

## A.1 Relation of Parametric UMAP to modularity clustering

The objective function of Parametric UMAP is loosely related to modularity clustering [15, 6]. The modularity of a clustering of a weighted graph is given by

$$
Q = \frac{1}{2\tilde{\mu}_{tot}}\sum_{i,j=1}^n\left(\tilde{w}_{ij}-\frac{\tilde{d}_i\tilde{d}_j}{2\tilde{\mu}_{tot}}\right)\cdot\delta(c_i,c_j),
\tag{25}
$$

where in accordance to our notation $\tilde{w}_{ij}$ is the weight of edge $ij$, $\tilde{d}_i = \sum_{j=1}^n \tilde{w}_{ij}$ is the degree of node $i$, $\tilde{\mu}_{tot} = \frac{1}{2}\sum_{i,j=1}^n \tilde{w}_{ij}$ is the total weight of the graph, $c_i$ is the cluster of node $i$ and finally $\delta$ is the Kronecker delta function. Modularity measures how much larger the edge weight within the clusters, $\sum_{i,j=1}^n \tilde{w}_{ij}\cdot\delta(c_i,c_j)$, is than the amount expected under the configuration model, $\sum_{i,j=1}^n \tilde{d}_i\tilde{d}_j/(2\tilde{\mu}_{tot})\cdot\delta(c_i,c_j)$. The $NP$-hard objective of modularity clustering is to find a clustering

with maximal modularity.

To relate this to the objective of Parametric UMAP, we observe that while $\delta(c_i, c_j)$ measures whether nodes $i$ and $j$ are in the same cluster, $\log(\nu_{ij})$ is maximal if the similarity of $e_i$ and $e_j$ in embedding space is high. Thus, both quantities encode whether in the output (a clustering or an embedding) $i$ and $j$ are deemed similar. Further, exchanging $\log(1 - \nu_{ij})$ with $-1 - \log(\nu_{ij})$ we can rewrite the negative of Parametric UMAP's loss as

$$\frac{1}{2(m+1)\mu_{\text{tot}}} \sum_{i,j=1}^{n} \left( \mu_{ij} \cdot \log(\nu_{ij}) + m\frac{b-1}{b}\frac{d_i d_j}{2\mu_{\text{tot}}} \cdot (-1 - \log(\nu_{ij})) \right) \tag{26}$$

$$= \frac{1}{2(m+1)\mu_{\text{tot}}} \sum_{i,j=1}^{n} \left( \left( \mu_{ij} - m\frac{b-1}{b}\frac{d_i d_j}{2\mu_{\text{tot}}} \right) \cdot \log(\nu_{ij}) \right) - \frac{1}{2(m+1)\mu_{\text{tot}}} \sum_{i,j=1}^{n} \left( m\frac{b-1}{b}\frac{d_i d_j}{2\mu_{\text{tot}}} \right) \tag{27}$$

$$= \frac{1}{2(m+1)\mu_{\text{tot}}} \sum_{i,j=1}^{n} \left( \left( \mu_{ij} - m\frac{b-1}{b}\frac{d_i d_j}{2\mu_{\text{tot}}} \right) \cdot \log(\nu_{ij}) \right) - \frac{m}{m+1}\frac{b-1}{b}. \tag{28}$$

In the last line we computed $\sum_{i,j} d_i d_j = 4\mu_{tot}^2$. As the last summand does not depend on $\nu_{ij}$, we see that minimizing Parametric UMAP's loss loosely corresponds to maximizing a quantity similar to modularity. Parametric UMAP's negative sampling is uniform from a batch that is itself sampled according to $\mu_{ij}$. The above computation shows that this yields a weighing of attractive and repulsive terms akin to modularity clustering. We leave a more in depth exploration of this connection for future work.

---

**Algorithm 2:** Parametric UMAP's sampling based optimization

**input** : high-dimensional similarities $\mu_{ij}$, negative sample rate $m$, number of epochs $T$, learning rate $\alpha$, embedding network $f_\theta$, batch size $b$

**output** : final embeddings $e_i$

1 **for** $\tau = 0$ **to** $T$ **do**
2     *Assemble batch*
3     $B_h, B_t = [], []$    // Initialize empty mini-batches for heads and tails
4     **for** $\beta = 1$ **to** $b$ **do**   // Sample edge by input similarity and add to batch
5        $ij \sim \text{Categorical}(\{1, \ldots, n\}^2, \{\frac{\mu_{rs}}{2\mu_{\text{tot}}}\}_{r,s=1,\ldots,n})$
6        $B_h.\text{append}(f_\theta(x_i))$
7        $B_t.\text{append}(f_\theta(x_j))$
8     *Compute loss*
9     $l = 0$
10     **for** $\beta = 1$ **to** $b$ **do**           // Add attractive loss for sampled edges
11        $l = l + \mathcal{L}^a(B_h[\beta], B_t[\beta])$
12     $\pi \sim \text{Uniform}(\text{permutations of } \{1, \ldots, m \cdot b\})$
13     **for** $\beta = 1$ **to** $mb$ **do**      // Add repulsive loss between negative samples
14        $l = l + \mathcal{L}^r(mB_h[\beta], mB_t[\pi(\beta)])$          // $mB$ repeats $B$ $m$ times
15     $l = \frac{l}{(m+1)b}$
16     *Update parameters*
17     $\theta = \theta - \alpha \cdot \nabla_\theta l$
18 **return** $f_\theta(x_1), \ldots, f_\theta(x_n)$

---

# B   UMAP degree distributions

Before symmetrization, the degree of each node $\vec{d_i} = \sum_{j=1}^{n} \mu_{i \to j}$ equals $\log_2(k)$ due to UMAP's uniformity assumption. For UMAP's default value of $k = 15$ this is $\approx 3.9$. For the C. elegans dataset we used $k = 30$ in which case $\log_2(30) \approx 4.9$. Symmetrizing changes the degree in a dataset-dependent way. Since $\max(a, b) \leq a + b - ab$ for $a, b \in [0, 1]$, the symmetric degrees $d_i = \sum_{j=1}^{n} \mu_{ij}$ are lower bounded by $\log_2(k)$. Empirically, we find that the degree distribution is fairly peaked close to this lower bound, see Figure 5.

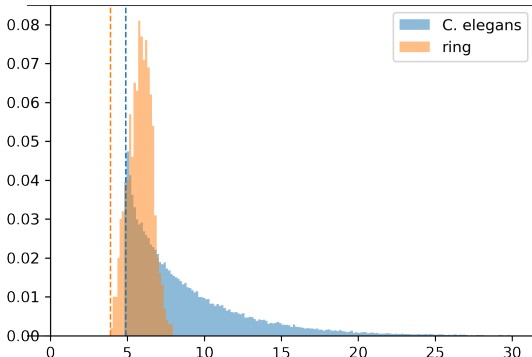

Figure 5: Histogram over the UMAP degree distributions for the toy ring and the C. elegans datasets. Both distributions are fairly peaked close to their lower bounds $\log_2(k)$, highlighted as dashed lines.

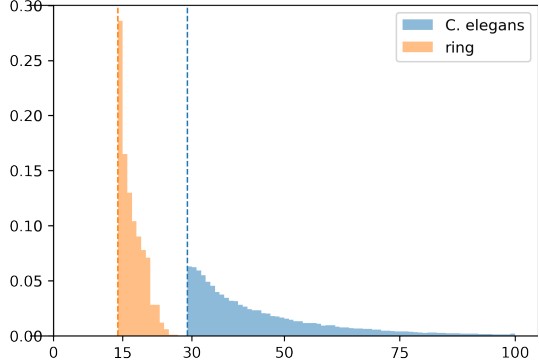

Figure 6: Histogram over the degree distribution in the shared $k$NN graph for the toy ring and the C. elegans datasets. Both distributions are fairly peaked close to their lower bounds $k - 1$, highlighted as dashed lines. Since UMAP's implementation considers a point its first nearest neighbor, but the $\mu_{ii}$ are set to zero, the degree is one lower than the intended number of nearest neighbors $k$.

In the shared $k$NN graph each node has degree at least $k$. Empirically, the degree distribution is fairly peaked at this lower bound, see Figure 6.

## C  Computing the expected gradient of UMAP's optimization procedure

In this appendix, we elaborate the computation that leads to equation (8):

$$\mathbb{E}\left(g_i^t\right) = \mathbb{E}\left(\sum_{j=1}^{n}\left(X_{ij}^t \cdot \frac{\partial \mathcal{L}_{ij}^a}{\partial e_i} + X_{ji}^t \cdot \frac{\partial \mathcal{L}_{ji}^a}{\partial e_i} + X_{ij}^t \cdot \sum_{s=1}^{n} Y_{ij,s}^t \cdot \frac{\partial \mathcal{L}_{is}^r}{\partial e_i}\right)\right)$$

$$= \sum_{j=1}^{n}\left(\mathbb{E}(X_{ij}^t) \cdot \frac{\partial \mathcal{L}_{ij}^a}{\partial e_i} + \mathbb{E}(X_{ji}^t) \cdot \frac{\partial \mathcal{L}_{ji}^a}{\partial e_i} + \sum_{s=1}^{n} \mathbb{E}(X_{ij}^t Y_{ij,s}^t) \cdot \frac{\partial \mathcal{L}_{is}^r}{\partial e_i}\right)$$

$$= \sum_{j=1}^{n}\left(\mu_{ij} \cdot \frac{\partial \mathcal{L}_{ij}^a}{\partial e_i} + \mu_{ji} \cdot \frac{\partial \mathcal{L}_{ji}^a}{\partial e_i}\right) + \sum_{s=1}^{n}\sum_{j=1}^{n} \frac{\mu_{ij} m}{n} \cdot \frac{\partial \mathcal{L}_{is}^r}{\partial e_i}$$

$$= 2\sum_{j=1}^{n}\left(\mu_{ij} \cdot \frac{\partial \mathcal{L}_{ij}^a}{\partial e_i} + \frac{d_i m}{2n} \cdot \frac{\partial \mathcal{L}_{ij}^r}{\partial e_i}\right). \tag{29}$$

From line 2 to 3, we computed $\mathbb{E}(X_{ij}^t Y_{ij,s}^t) = \mathbb{E}_{X_{ij}^t}\left(X_{ij}^t \cdot \mathbb{E}(Y_{ij,s}^t | X_{ij}^t)\right) = \frac{\mu_{ij} m}{n}$ and from line 3 to 4 we used the symmetry of $\mu_{ij}$ and $\mathcal{L}_{ij}^a$ and collected the high-dimensional similarities $\sum_j \mu_{ij}$ into the degree $d_i$.

# D UMAP's update rule has no objective function

In this appendix, we show that the expected update in UMAP's optimization scheme does not correspond to any objective function. Recall that the expected update of an embedding $e_i$ in UMAP's optimization scheme (8) is

$$\mathbb{E}\left(g_i^t\right) = 2\sum_{j=1}^{n}\left(\mu_{ij}\cdot\frac{\partial\mathcal{L}_{ij}^a}{\partial e_i} + \frac{d_i m}{2n}\cdot\frac{\partial\mathcal{L}_{ij}^r}{\partial e_i}\right). \tag{30}$$

It is continuously differentiable unless two embedding points coincide. Therefore, if it had an antiderivative, that would be twice continuously differentiable at configurations where all embeddings are pairwise distinct and thus needs to have a symmetric Hessian at these points. However, we have

$$\frac{\partial\mathbb{E}\left(\frac{\partial\mathcal{L}^t}{\partial e_i}\right)}{\partial e_j} = 2\mu_{ij}\cdot\frac{\partial^2\mathcal{L}_{ij}^a}{\partial e_j\partial e_i} + \frac{d_i m}{2n}\cdot\frac{\partial\mathcal{L}_{ij}^r}{\partial e_j\partial e_i}$$

$$\frac{\partial\mathbb{E}\left(\frac{\partial\mathcal{L}^t}{\partial e_j}\right)}{\partial e_i} = 2\mu_{ij}\cdot\frac{\partial^2\mathcal{L}_{ij}^a}{\partial e_i\partial e_j} + \frac{d_j m}{2n}\cdot\frac{\partial\mathcal{L}_{ij}^r}{\partial e_i\partial e_j}. \tag{31}$$

Since $\mathcal{L}_{ij}^a$ and $\mathcal{L}_{ij}^r$ are themselves twice continuously differentiable, their second order partial derivatives are symmetric. But this makes the two expressions in equation (31) unequal unless $d_i$ equals $d_j$.

The problem is that negative samples themselves are not updated, see commented Algorithm 1 line 10. We suggest to remedy this by pushing the embedding of a negative sample $i$ away from the embedding node $e_j$, whenever $i$ was sampled as negative sample to some edge $jk$. This yields the gradient in equation (10) at epoch $t$.

# E Negative Sampling in LargeVis

Our analysis is also applicable to the visualization method LargeVis [18], which uses a very similar optimization scheme as UMAP, but with a slightly different negative sampling distribution. Its intended loss function is given in [18] as

$$-\sum_{ij\in\text{shared }k\text{NN graph}}\mu_{ij}\log(\nu_{ij}) - \sum_{ij\notin\text{shared }k\text{NN graph}}\gamma\log(1-\nu_{ij}), \tag{32}$$

with a constant $\gamma$, set to 7 per default in [18]. Different from UMAP, $\mu_{ij}$ and $\nu_{ij}$ are the high- and low-dimensional similarities from $t$-SNE [19]; in particular the $\mu_{ij}$ are only non-zero on the edges of the shared $k$NN graph. For optimization, Tang et al. [18] employ negative sampling and arrive at the following, different objective function

$$-\sum_{ij\in\text{shared }k\text{NN graph}}\mu_{ij}\left(\log(\nu_{ij}) + \sum_{s=1}^{m}\mathbb{E}_{j_s\sim P(a)}\big(\gamma\log(1-\nu_{ij_s})\big)\right), \tag{33}$$

with negative sample distribution $P(a) \propto d_a^{0.75}$. As done in our paper, we can compute the expectation and rearrange to arrive at a closed form loss function:

$$-\sum_{ij \in \text{shared } k\text{NN graph}} \mu_{ij}\left(\log(\nu_{ij}) + \sum_{s=1}^{m} \mathbb{E}_{j_s \sim P(a)}\big(\gamma \log(1 - \nu_{ij_s})\big)\right)$$

$$= -\frac{1}{2}\sum_{1 \leq i,j \leq n} \mu_{ij}\left(\log(\nu_{ij}) + \sum_{s=1}^{m} \mathbb{E}_{j_s \sim P(a)}\big(\gamma \log(1 - \nu_{ij_s})\big)\right)$$

$$= -\frac{1}{2}\sum_{1 \leq i,j \leq n} \mu_{ij}\left(\log(\nu_{ij}) + m\gamma \sum_{\alpha=1}^{n} \left(\frac{d_\alpha^{0.75}}{\sum_{l=1}^{n} d_l^{0.75}} \log(1 - \nu_{i\alpha})\right)\right)$$

$$= -\frac{1}{2}\sum_{1 \leq i,j \leq n} \mu_{ij} \log(\nu_{ij}) - \frac{m\gamma}{2}\sum_{1 \leq i,j \leq n} \left(\mu_{ij} \sum_{\alpha=1}^{n} \left(\frac{d_\alpha^{0.75}}{\sum_{l=1}^{n} d_l^{0.75}} \log(1 - \nu_{i\alpha})\right)\right)$$

$$= -\frac{1}{2}\sum_{1 \leq i,j \leq n} \mu_{ij} \log(\nu_{ij}) - \frac{m\gamma}{2}\sum_{1 \leq i,\alpha \leq n} \left(\frac{d_\alpha^{0.75}}{\sum_{l=1}^{n} d_l^{0.75}} \log(1 - \nu_{i\alpha}) \sum_{j=1}^{n} \mu_{ij}\right)$$

$$= -\frac{1}{2}\sum_{1 \leq i,j \leq n} \mu_{ij} \log(\nu_{ij}) - \frac{m\gamma}{2}\sum_{1 \leq i,\alpha \leq n} \left(\frac{d_i d_\alpha^{0.75}}{\sum_{l=1}^{n} d_l^{0.75}} \log(1 - \nu_{i\alpha})\right)$$

$$= -\frac{1}{2}\sum_{1 \leq i,j \leq n} \mu_{ij} \log(\nu_{ij}) - \frac{1}{2}\sum_{1 \leq i,\alpha \leq n} \left(\frac{m\gamma(d_i d_\alpha)^{0.75}(d_i^{0.25} + d_\alpha^{0.25})}{2\sum_{l=1}^{n} d_l^{0.75}} \log(1 - \nu_{i\alpha})\right)$$

$$= -\frac{1}{2}\sum_{1 \leq i,j \leq n} \mu_{ij} \log(\nu_{ij}) - \frac{1}{2}\sum_{1 \leq i,j \leq n} \left(\frac{m\gamma(d_i d_j)^{0.75}(d_i^{0.25} + d_j^{0.25})}{2\sum_{l=1}^{n} d_l^{0.75}} \log(1 - \nu_{ij})\right)$$

$$= -\sum_{1 \leq i < j \leq n} \left(\mu_{ij} \log(\nu_{ij}) + \frac{m\gamma(d_i d_j)^{0.75}(d_i^{0.25} + d_j^{0.25})}{2\sum_{l=1}^{n} d_l^{0.75}} \log(1 - \nu_{ij})\right). \tag{34}$$

First, we used the fact that $\mu_{ij}$ is zero outside of the $k$NN graph, in particular that $\mu_{ii} = 0$. Second, we computed the expectation

$$\mathbb{E}_{j_s \sim P(a)}\big(\log(1 - \nu_{ij_s})\big) = \sum_{\alpha=1}^{n} \left(\frac{d_\alpha^{0.75}}{\sum_{l=1}^{n} d_l^{0.75}} \log(1 - \nu_{i\alpha})\right). \tag{35}$$

At this point, we use the same convention for the undefined term $\log(1 - \nu_{ii}) = \log(0)$ as described in the beginning of Appendix F. Third and fourth, we rearranged the summations. Fifth, we computed the degree $d_i$, and sixth, we used the symmetry $\nu_{i\alpha} = \nu_{\alpha i}$. Seventh, we relabelled $\alpha$ to $j$, and finally, we rearranged the summation one last time using both $\mu_{ii} = 0$ and the convention for $\log(0)$.

LargeVis does not motivate its loss function as a sum of binary cross-entropy losses for each edge, but instead as attraction on edges of the $k$NN graph plus constant repulsion on all non-$k$NN edges. Nevertheless, the negative sampling based optimization turns this into a sum of non-normalized binary cross-entropy losses as for UMAP. While we have not computed values of the repulsion pre-factor empirically, we strongly believe that it is very small, since the denominator contains a sum over the entire dataset. In this case, what we called "target similarities" would look binary for LargeVis as well. This puts LargeVis in even closer proximity to UMAP as the different choice of input similarity $\mu_{ij}$ in both methods matters little and helps to explain why LargeVis and UMAP embeddings often look very similar.

## F  Implementation details

To deal with the quadratic complexity when computing the dense low-dimensional similarities $\nu_{ij}$, we used the Python package PyKeOps [5] that parallelizes the computations on the GPU.

The repulsive loss $\mathcal{L}_{ij}^{r}$ is undefined for $i = j$. However, the implemented gradient update treats $\frac{\partial \mathcal{L}_{ii}^{r}}{\partial e_i}$ as zero. As $\mu_{ii} = 0$, we can thus safely replace $2\sum_{j=1}^{n}$ by $2\sum_{j=1, i \neq j}^{n}$ in formulae for expected

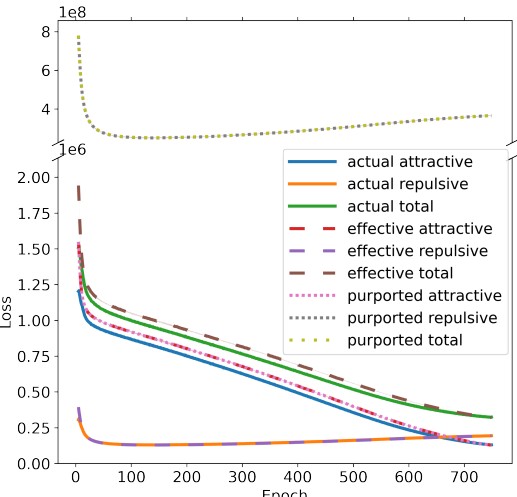

Figure 7: Same as Figure 3, but actual losses are computed with the embeddings at the time of update not with the embeddings after the full epoch as all other losses.

updates. Moreover, we may treat $\mathcal{L}_{ii}^r$ as a constant, so that its inclusion in the loss does not matter for the optimization. Together with the symmetry of $\mu_{ij}$, $\mathcal{L}_{ij}^a$ and $\mathcal{L}_{ij}^r$, we can thus also replace $\sum_{i,j=1}^n$ with $2\sum_{1\leq i<j\leq n}$ in formulae for loss functions. When logging losses, we include the $\mathcal{L}_{ii}^r$ terms. To guard us against numerical instabilities from $\log$, we always use $\log(\min(x + 0.0001, 1))$ instead of $\log(x)$.

When computing the various loss terms for UMAP, we use the embeddings after each full epoch. The embeddings in UMAP are updated as soon as the an incident edge is sampled. Thus, an embedding might be updated several times during an epoch and gradient computations always use the current embedding, which might differ slightly from the embedding after the full epoch. Logging the loss given the embeddings at the time of each individual update yields as slightly lower attractive loss term, see Figure 7.

Our description of UMAP's implementation is based on the original paper [13] and version 0.5.0 of the umap-learn package.[7]

Our code is publicly available at `https://github.com/hci-unihd/UMAPs-true-loss`.

### F.1 Stability

Whenever we report loss values, we computed the average over seven runs and give an uncertainty of one standard deviation. Sources of randomness are in the approximate $k$NN computation via nearest neighbor descent [7], the Gaussian noise added to the Laplacian Eigenmap initialization, the negative sampling and the sampling of the toy data. Note that the sampling of attractive pairs is implemented deterministically and includes an edge $ij$ every $\max_{ab} \mu_{ab}/\mu_{ij}$-th epoch. We find that the deviation in the loss values is very small across different runs. In fact, as the standard deviation is barely visible in Figure 3, we include the same figure but with shaded areas corresponding to ten standard deviations in Figure 8. Nevertheless, the visual effect of different random seeds can be noticeable as depicted in Figure 9.

### F.2 Compute

We ran all our experiments on a machine with 20 "Intel(R) Xeon(R) Silver 4114 CPU @ 2.20GHz" CPUs and six "Nvidia GeForce GTX 1080 Ti" GPUs. We only ever used a single GPU and solely for computing the effective and purported losses $\tilde{\mathcal{L}}$ (eq. (12)) and $\mathcal{L}$ (eq. (6)). Table 2 shows the run times for the main experiments averaged over 7 runs. Uncertainties indicate one standard deviation. Logging the losses during optimization quintuples UMAP's run time on the C. elegans dataset. This

---

[7]https://github.com/lmcinnes/umap

Table 2: Run times of key experiments averaged over seven runs with one standard deviation and number of runs of similar experiments needed to reproduce the paper

| Experiment | Run rime [s] | Number of runs |
|---|---|---|
| C. elegans without loss logging (Fig. 14) | $520 \pm 14$ | 15 |
| C. elegans with loss logging (Fig. 2) | $2427 \pm 7$ | 21 |
| PBMC (Fig. 17) | $2087 \pm 7$ | 7 |
| Lung Cancer (Fig. 18) | $1491 \pm 4$ | 7 |
| CIFAR-10 (Fig. 19) | $302 \pm 0.3$ | 7 |
| Toy ring (Fig. 1b) | $48 \pm 0.01$ | 12 |
| Multiple rings (Fig. 10) | $1726 \pm 17$ | 7 |
| Toy ring of variable size (Fig. 14) | $1647 \pm 13$ | 7 |
| Toy ring with dense $\mu_{ij}$'s (Fig. 1d) | $1040 \pm 2$ | 10 |
| Toy ring with dense $\mu_{ij}$'s and variable size (Fig. 15)[8] | $52349 \pm 224$ | 2 |

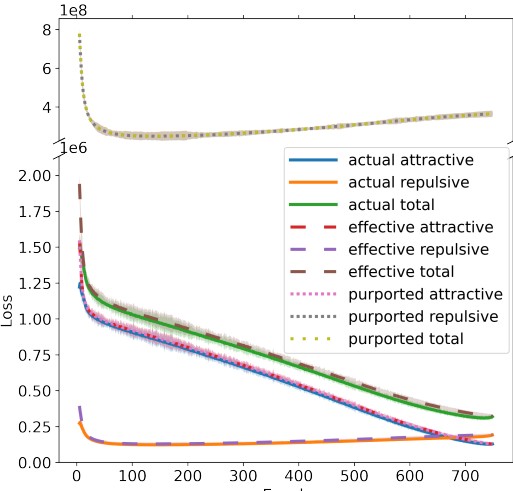

Figure 8: Same as Figure 3 but here the shaded region corresponds to ten standard deviations.

is due to the quadratic complexity of evaluating the effective and purported loss functions. But with our GPU implementation this longer run time is still easily manageable for the reasonably large real world C. elegans dataset. The toy ring experiments with a dense and thus much larger input graph take about 25 times longer than with the normal, sparse input similarities.

We estimate the total compute by adding the run times of the experiments necessary to reproduce the paper. The number of comparable experiments needed to reproduce the paper is given in Table 2. The total run time amounts to about 63 hours.

## G Quantitative metrics

While it is natural to wonder about metrics that quantify how faithful and thus "good" a low-dimensional visualization is, there is no consensus on which metrics align best with the subjective, human impression of a "descent" visualization. Becht et al. [2] employ the Pearson correlation between all pairwise distances in high- and low-dimensional space for a subsample of the dataset (for efficiency). Kobak and Berens [9] use the Spearman correlation instead and considers it a "global" measure for the faithfulness of the embedding. As "local" measure Kobak and Berens [9] compute the average share of $k$ nearest neighbors of a point in high dimension that are also $k$ nearest neighbors in the low-dimensional embedding. They use $k = 10$. We have computed these measures for the two-dimensional visualizations of the C.elegans dataset with PCA, UMAP and the UMAP version with inverted weights, see Figure 2. We used subsamples of size 10000 from the 86024 cells in the

---

[8]Only two runs were measured due to the long run time.

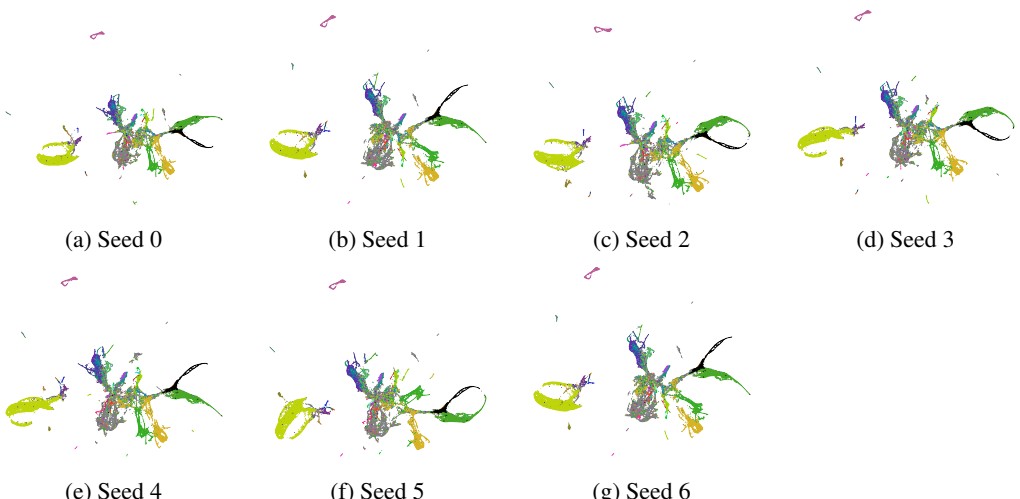

<table>
<tr><td>(a) Seed 0</td><td>(b) Seed 1</td><td>(c) Seed 2</td><td>(d) Seed 3</td></tr>
<tr><td>(e) Seed 4</td><td>(f) Seed 5</td><td>(g) Seed 6</td><td></td></tr>
</table>

Figure 9: Visualizations of the C. elegans dataset with UMAP for seven different random seeds. While the losses vary very little, see Figures 3 and 8, the visualizations show significant differences, such as closed versus open loops and placement of subgroups. All plots were subjectively flipped and rotated by multiples of $\pi/2$ to ease a visual comparison.

Table 3: Quantitative measures for visualizations of the C.elegans dataset with PCA, UMAP with inverted positive input weights and original UMAP. Arrows indicate whether higher or lower is better. Best method in **bold**.

| Metric | PCA | UMAP inv | UMAP |
|---|---|---|---|
| Pearson's r $\uparrow$ | $0.577 \pm 0.003$ | $\mathbf{0.628 \pm 0.005}$ | $0.562 \pm 0.008$ |
| Spearman's r $\uparrow$ | $\mathbf{0.611 \pm 0.003}$ | $0.607 \pm 0.006$ | $0.551 \pm 0.011$ |
| kNN accuracy ($k = 10$) $\uparrow$ | $0.006 \pm 0.000$ | $0.079 \pm 0.001$ | $\mathbf{0.156 \pm 0.001}$ |
| kNN accuracy ($k = 30$) $\uparrow$ | $0.013 \pm 0.000$ | $0.185 \pm 0.001$ | $\mathbf{0.256 \pm 0.001}$ |
| KL divergence pos $\downarrow$ | $0.693 \pm 0.000$ | $0.577 \pm 0.002$ | $\mathbf{0.549 \pm 0.002}$ |
| KL divergence $\downarrow$ | $6.46 \pm 0.00$ | $5.13 \pm 0.01$ | $\mathbf{4.92 \pm 0.00}$ |
| purported UMAP loss [$10^8$] $\downarrow$ | $5.37 \pm 0.00$ | $4.32 \pm 0.03$ | $\mathbf{3.64 \pm 0.02}$ |
| true UMAP loss [$10^5$] $\downarrow$ | $12.4 \pm 0.00$ | $3.85 \pm 0.02$ | $\mathbf{3.21 \pm 0.02}$ |

C. elegans dataset for the correlation measures. For the $k$NN accuracy we used $k = 10$ as in [9] as well as $k = 30$, because on the C. elegans dataset we used 30 neighbors in UMAP's approximate $k$NN graph computation. In addition, we report Kullback-Leibler divergences between the input and embedding similarities as this is the objective function in $t$-SNE [20, 19]. Note that different from $t$-SNE, in UMAP neither the input nor the embedding similarities are normalized over all pairs of points. We report two versions of the KL diveregence. In both, we normalize the input similarities. In the first, we normalize the embedding similarities only with respect to the pairs of points that have positive input similarity. In the second, we normalize with respect to all pairs of points. Finally, we also compute the loss values of UMAP's purported and true loss function. Both in the KL divergences and the two losses, we take the original UMAP input similarities as reference. We report the mean and standard deviation over 7 runs in Table 3.

As previously observed by Kobak and Berens [9], PCA is good at preserving the global structure as indicated by the high correlation coefficients. However, we see very high Pearson correlation for the inverted UMAP setting. On the other hand, PCA is much worse than both UMAP methods at preserving the local neighborhood structure. This is unsurprising as UMAP tries to reproduce local similarities in low-dimensional space, which are positive only on the $k$NN edges. Our binarization analysis shows that UMAP essentially just uses the binary high-dimensional $k$NN structure to guide its low-dimensional embedding. The inverted UMAP setting still does a lot better than PCA on the $k$NN accuracies, but not quite as well as the original UMAP, presumably because the effect of inverting the positive input similarities is not completely effaced by the binarization.

The two KL divergences and the true UMAP loss paint a similar picture. The inverted UMAP version is nearly as good as the original while PCA is much worse. On the purported UMAP loss, the ranking is the same, but the difference between PCA and the two UMAP versions is less extreme.

The results of the $k$NN accuracies, the KL divergences and the true UMAP loss are in accordance with the visual quality of the embeddings and might suggest that our effective UMAP loss aligns well with what one might call a "good" embedding.

## H   UMAP's dependence on the dataset size

UMAP's true loss function (12) depends on the dataset, in particular on its size $n$. As discussed in the paper, the reduction of repulsion, the binarization and the over-contraction will all be stronger for larger datasets. This appendix discusses this observation in more detail and explores the effect empirically.

### H.1   Multiple ring experiments

We applied UMAP to datasets containing multiple 2D rings of 1000 points each. The rings were spaced so that no point in one ring had any of its $k$ nearest neighbors in another ring. As the input similarities $\mu_{ij}$ only depend on the distances to the $k$ nearest neighbors, this ensures that the input similarities for points from one ring are the same as if the dataset consisted only of that ring. The input similarities between points in different rings are all zero. We varied the number of rings from 1 to 20. We formed the datasets by iteratively adding rings, so the datasets with fewer rings are subsets of those with more rings. To ensure convergence, we ran the UMAP optimization for 10000 epochs as for the single ring experiments in Figure 1. Moreover, we initialized the optimization at the original layout. Other hyperparameters were kept at their default values. The results can be found in Figure 10. With this setup, one might expect that the embedding of the first ring would look similar for all datasets. However, what we observe is that the embedding of the rings gets crisper the more rings the dataset contains. The repulsion weight in our effective loss function (12) depends on the degrees of the points, the negative sample rate and the dataset size $n$. Since the input similarities are the same as in individual rings by construction, the degrees of each point are independent of the presence of other rings. But the dataset size of course increases with the number of rings, so that the repulsion gets smaller for the larger datasets. Viewed from the point of the target similarities (15), the binarization gets stronger. This explains why we see more over-contraction. In Figure 11, we show histograms for input, target and embedding similarities (as graphs for visual clarity), which confirm that the input similarities do not change fundamentally, but the target and thus the embedding similarities get more skewed towards one as the number of rings increases. We can also give a more concrete explanation for the reduced repulsion. The edges for attraction are sampled equally frequently, independent of the number of rings. For each such sampled attractive edge, $m$ negative samples are generated. But these may come from any of the rings in the dataset. So if there are more rings, it gets more likely that a negative sample comes from a different ring. Consequently, it gets less likely that it comes from the same ring. So repulsion within the ring is decreased while the attraction within the ring remains the same, explaining the more contracted embeddings.

### H.2   Varying the sample size of a single ring

In this set of experiments, we computed UMAP embeddings datasets consisting of a varying number of points sampled from a 2D ring. We sampled between 100 and 10000 points. As for the case of multiple rings above, our analysis predicts that the larger the dataset, the stronger the reduction in repulsion, the binarization and the over-contraction. Indeed, this is what we see in Figure 12, where we show the distributions of input, target and embedding similarities for pairs of points with positive input similarity. Because of UMAP's uniformity assumption, the size of the input similarities depends on the relative distances of a point to its $k$ nearest neighbors. Since the points are always sampled uniformly from the ring, we can expect that the distribution of input similarities does not change qualitatively as we increase the sample size, which is confirmed in Figure 12a. For the target similarities, a larger number of points implies that they should be more skewed towards one, as we see in Figure 12b. Note that for the smallest datasets, speaking of binarization is not quite justified. E.g., the dataset with 100 points has a typical repulsion strength of $\frac{(d_i+d_j)m}{2n} \approx 0.2$, so that the largest

possible target similarity is $1/1.2 \approx 0.83$. Finally, the embedding similarities also get skewed more towards one as the dataset gets larger, indicating increasing over-contraction.

As usual, we optimized the UMAP embeddings for 10000 epochs, initialized at the original positions and kept all other hyperparameters at their defaults. The resulting embeddings can be found in Figure 14. While our theory predicts increasing over-contraction as the dataset gets larger, corroborated by the similarity distributions in Figure 12, the embedding of the dataset with 500 points appears to have the smallest ring width, while the embeddings of the dataset with 100 or of those with more than 500 points seem to have larger ring widths. In particular, for the dataset of 10000 points the apparent ring width is nearly as large as in the input data. But this is not actually at odds with our predictions: An over-contracted ring width is only one way in which over-contraction can manifest. In the zoomed-in segments of the ring embeddings in Figure 14, we see that the structure of points within the embedding is far from uniform: Points cluster or form lines, much like in Figure 22, where we apply UMAP to uniform data. So the predicted and measured over-contraction is present, but on a smaller scale than the ring width, so that it is not immediately visible from the global embedding. Note these are spurious artifacts of UMAP's optimization procedure as the original data is uniform.

We conjecture that the reason for whether one observes over-contraction on a global scale has to do with the relative size of the local neighborhoods and the width of the ring. If the local neighborhood is large relative to the width of the ring, it is more likely to be contracted to a line. If it is small, the over-contraction happens within the width of the ring. Another way of looking at this phenomenon is that for fixed $k$ but increasing sample size of the toy ring, the distance to the $k$-th nearest neighbor and thus the radius of the local neighborhood decrease relative to the fixed width of the whole ring. We computed the mean distance to the $k$-th nearest neighbor over all data points and plotted the results in Figure 13. At 500 data points, when the width of the embedding has nearly reached 0, the mean $k$-th nearest neighbor radius is about 0.54. Hence, the local neighborhoods of some points span the width of the ring. At 1000 points, when the embedding ring width is increasing again, the average $k$-th nearest neighbor radius is only 0.35, smaller than the original ring width of 1. We note for completeness that for some toy rings with few points, the ring structure was torn up to a line or even a set of clusters (not shown). For the dataset with only 100 points, the mean distance to the $k$-th nearest neighbor is about 2, but the small number of data points reduces the repulsion strength not enough for the embedding to become visibly over-contracted.

Since both applying UMAP on an independent subset or a down- or up-sampled version of a dataset are common, we expect the resulting artifacts to appear often in practice.

### H.3 Varying the sample size for dense input similarities

We also computed UMAP embeddings based on dense input similarities $\mu_{ij} = \phi(||x_i - x_j||)$ for rings with varying number of samples. Here, the analysis is different than for the typical, $k$NN based input similarities. As the dataset size increases, each point is similar to more points. Therefore, the degree $d_i$ for a point $i$ scales with the dataset size and the effective repulsive pre-factor $\frac{(d_i+d_j)m}{2n}$ does not decrease with the dataset size but remains approximately constant. We show the linear scaling of the average degree with the dataset size in Figure 16. Hence, we do not expect any qualitative difference in this setting as we vary the number of points in the ring. Indeed, experiments in which we changed the number of points of the ring from 100 to 5000 confirm this prediction, see Figure 15.

### H.4 Discussion of the influence of $m$, $k$ and $n$

The effective repulsion weight $\frac{(d_i+d_j)m}{2n}$ and thus the transformation from input to target similarities depends on the degrees of nodes $i$ and $j$, the negative sample rate $m$ and the dataset size $n$. As discussed in Section 3 and Appendix B, the degree $d_i = \sum_{j=1}^{n} \mu_{ij}$ is typically close to $\log_2(k)$. Plugging this into equation (15) shows that the target similarities of UMAP's implementation are roughly

$$\nu_{ij}^* \approx \frac{\mu_{ij}}{\mu_{ij} + \log_2(k)m/n}. \tag{36}$$

This shows how the relation between target and high-dimensional similarities depends on $k$, $m$ and $n$. Note that there are no parameter settings for $m$ and $k$ that allow the (near) exact reconstruction of the whole value range of input similarities $\mu_{ij}$'s for a fixed $n$ as this would require

$$\log_2(k)m/n = 1 - \mu_{ij} \forall i, j. \tag{37}$$

For instance, we can see in Figure 4 that most non-zero high-dimensional similarities are either $1$ or close to $0.05$ for the C. elegans dataset with $k = 30$ and $n = 86024$. To preserve the $\mu_{ij}$'s that equal $1$, we would need

$$m = (1 - 1) \cdot 86024/\log_2(30) = 0, \tag{38}$$

while to preserve the $\mu_{ij}$'s near $0.05$, we would need

$$m = (1 - 0.05) \cdot 86024/\log_2(30) \approx 16500 \tag{39}$$

negative samples, which would be prohibitively slow computationally. Changing $k$ would additionally change the algorithm's notion of locality, which has implications on both speed and appearance.

As another example, consider UMAP's default setting of $k = 15$ and $m = 5$. Then already for $n = 500$ each input similarity $\mu_{ij} > 0.2$ would be mapped to a target similarity

$$\nu_{ij}^* \gtrapprox \frac{0.2}{0.2 + \frac{5 \log_2(15)}{500}} \approx 0.83. \tag{40}$$

So even for relatively small datasets the over-contraction can be strong, even if the target similarities are not necessary binary.

It is tempting to adapt $m$ and $k$ to the dataset size, so that $\log_2(k)m/n$ remains constant and thus UMAP becomes less dependent on the dataset size. But as mentioned above, increasing $m$ with $n$ makes UMAP's run time quadratic in $n$ and increasing $\log_2(k)$ with $n$ makes UMAP's scale exponentially with $n$ and would also change the notion of locality. The most severe problem that we see with keeping the repulsion pre-factor $\frac{(d_i + d_j)m}{2n}$ constant is that this means that the total amount of repulsion acting on a single embedding point scales with the dataset size while the total amount of attraction remains constant. This way, we risk to arrive at diverged embeddings for sufficiently large datasets, as observed by Böhm et al. [4] and discussed in Section 7.2. While the inverse relation of the repulsion pre-factor with the dataset size can lead to over-contraction on the $k$NN graph edges, it also counteracts the quadratic number of non-$k$NN graph edges.

# I   Additional datasets

## I.1   PBMC dataset

In this section we analyze the UMAP embedding of the scRNA-seq PMBC dataset of [22]. The dataset consists of gene expression measurements for $68551$ peripheral blood mononuclear cells. We used the $50$-dimensional pre-processed version of the dataset from [14].[9] We used the same hyperparameters as for the C. elegans dataset. In other words, we used the cosine distance in input space, worked with $k = 30$ nearest neighbors and optimized for $750$ epochs. The results can be found in Figure 17. We see over-contraction in the embedding, that only our effective loss matches the measured sampling based loss and that the embedding similarities approximate the nearly binary target similarities.

## I.2   Lung cancer dataset

In this section we analyze the UMAP embedding of the scRNA-seq dataset of [23] containing gene expression measurements for $48969$ lung cancer and immune cells. We used the $306$-dimensional pre-processed version of the dataset from [14].[10] We used the same hyperparameters as for the C. elegans dataset. In other words, we used the cosine distance in input space, worked with $k = 30$ nearest neighbors and optimized for $750$ epochs. The results can be found in Figure 18. We see over-contraction in the embedding, that only our effective loss matches the measured sampling based one and that the embedding similarities approximate the nearly binary target similarities.

---

[9]obtained from `http://cb.csail.mit.edu/cb/densvis/datasets/`. We informed the authors of our use of the dataset, which they license under CC BY-NC 2.0.

[10]obtained from `http://cb.csail.mit.edu/cb/densvis/datasets/`. We informed the authors of our use of the dataset, which they license under CC BY-NC 2.0.

## I.3 Resnet50 features of CIFAR-10

In this section, we corroborate our results on image data. More precisely, we use the CNN backbone of a Resnet50 [8] pretrained on ImageNet as feature extractor. With it we obtain 2048-dimensional image features for the CIFAR-10 dataset [11].[11] These high-dimensional features are then embedded to two dimensions via UMAP with default hyperparameters. The resulting embedding and the loss curves are depicted in Figure 19, which supports our theoretical predictions.

We used the pretrained Resnet50 from torchvision 0.8.2 and transformed the CIFAR-10 images as expected by the network. While ImageNet labels were used for pretraining the Resnet50, no CIFAR-10 labels were used for the feature extraction or the UMAP dimension reduction. For this reason, we used the training and test set of CIFAR-10 jointly.

---

[11]licensed under the MIT license, see `https://peltarion.com/knowledge-center/documentation/terms/dataset-licenses/cifar-10`

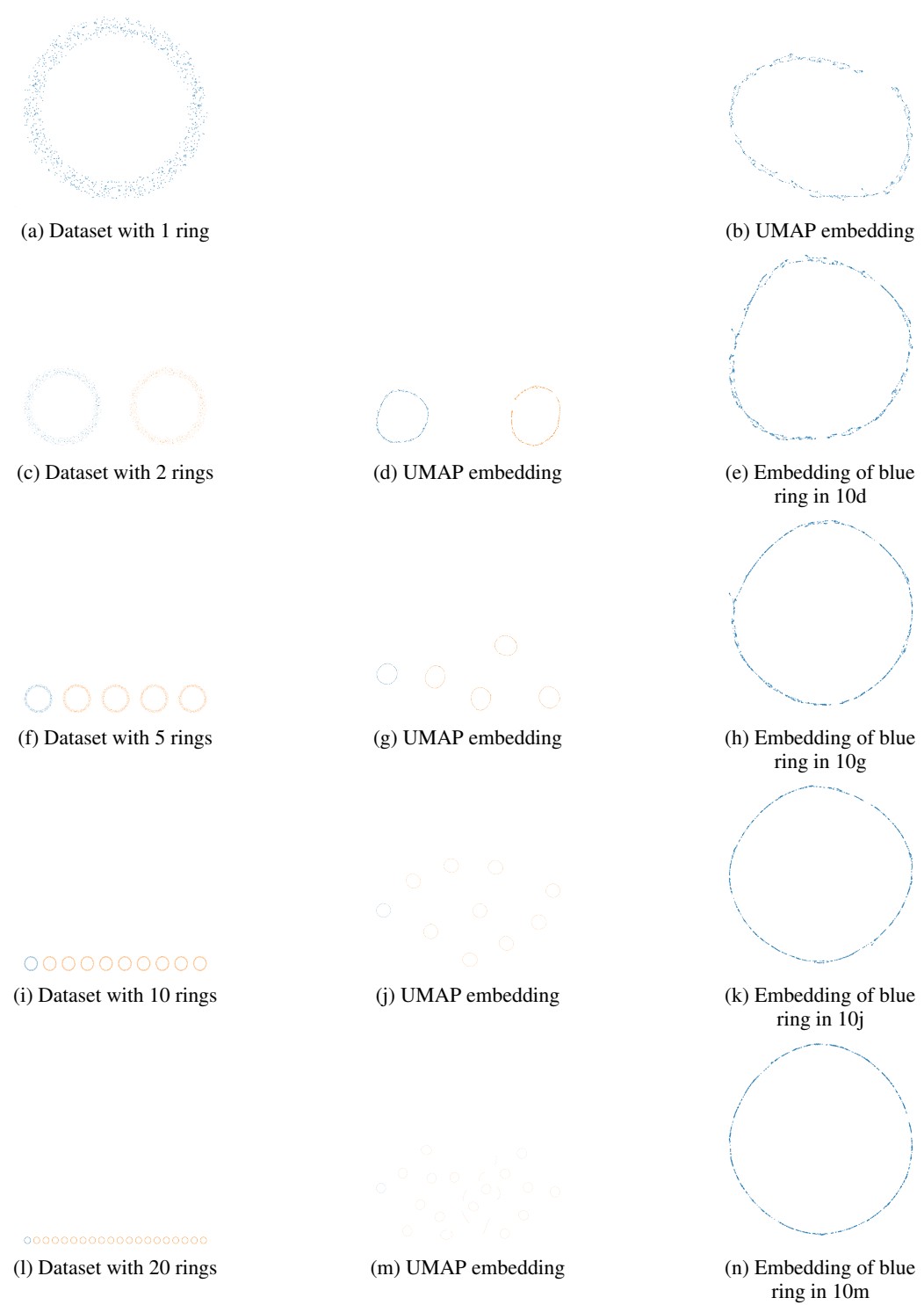

(a) Dataset with 1 ring

(b) UMAP embedding

(c) Dataset with 2 rings

(d) UMAP embedding

(e) Embedding of blue ring in 10d

(f) Dataset with 5 rings

(g) UMAP embedding

(h) Embedding of blue ring in 10g

(i) Dataset with 10 rings

(j) UMAP embedding

(k) Embedding of blue ring in 10j

(l) Dataset with 20 rings

(m) UMAP embedding

(n) Embedding of blue ring in 10m

Figure 10: UMAP's embedding of datasets with multiple rings. **Left column:** Datasets of multiple 2D rings à 1000 points per ring. Datasets with fewer points are subsets of those with more points. The blue ring is the same in all datasets. Larger dataset are plotted smaller for shortage of space. **Middle column:** UMAP embedding of the dataset on the left. The larger the dataset, the more contracted the ring width. Embeddings of larger datasets are plotted smaller for shortage of space. **Right column:** Embedding of the blue ring at the same size. The larger the dataset, the crisper the embedding becomes although the blue ring is the same in all datasets. Figure best viewed digitally.

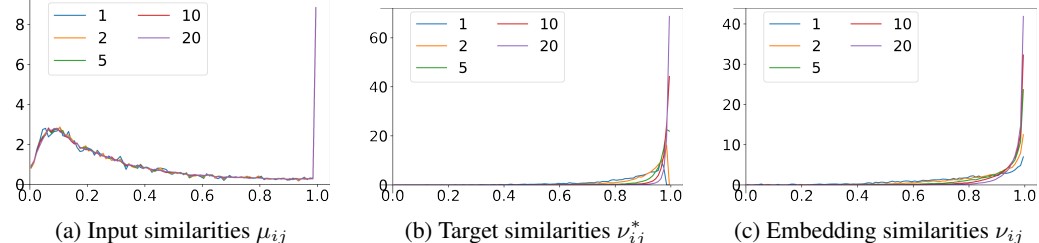

(a) Input similarities $\mu_{ij}$      (b) Target similarities $\nu_{ij}^*$      (c) Embedding similarities $\nu_{ij}$

Figure 11: Distributions of input ($\mu_{ij}$), target ($\nu_{ij}^*$) and embedding similarities ($\nu_{ij}$) for pairs with positive input similarity from datasets with a varying number of rings. All distributions are normalized to one. The legends indicate the number of rings in the dataset. **11a:** The high-dimensional similarities do not change qualitatively between the datasets. Their distribution only gets smoother as the dataset gets larger. **11b:** The larger the dataset, the stronger the binarization and thus the more the distributions of target similarities are skewed to one . **11c:** The larger the dataset, the stronger the over-contraction and thus the more the distributions of embedding similarities are skewed to one.

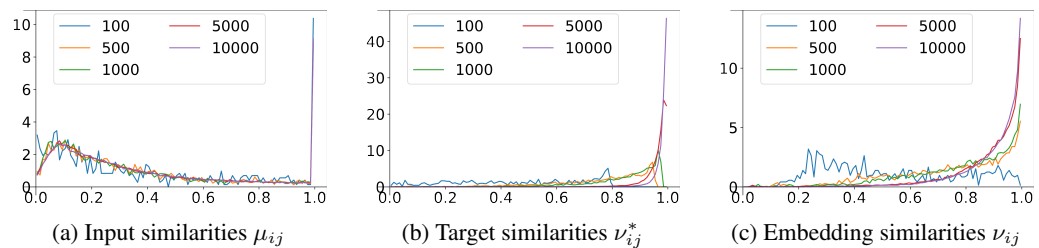

(a) Input similarities $\mu_{ij}$      (b) Target similarities $\nu_{ij}^*$      (c) Embedding similarities $\nu_{ij}$

Figure 12: Distributions of input ($\mu_{ij}$), target ($\nu_{ij}^*$) and embedding similarities ($\nu_{ij}$) for pairs of points with positive input similarity from single ring datasets with a varying number of points. All distributions are normalized to one. The legends indicate the number of sample points of the ring. **11a:** The high-dimensional similarities do not change qualitatively between the datasets. Their distribution only gets smoother as the dataset gets larger. **11b:** The larger the dataset, the stronger the binarization and thus the more the distributions of target similarities are skewed to one. In particular, we see that for smaller datasets, the target similarities are bound away from one by equation (15). **11c:** The larger the dataset, the stronger the over-contraction and thus the more the distributions of embedding similarities are skewed to one.

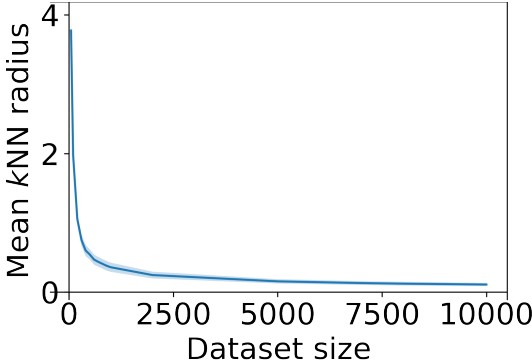

Figure 13: Average distance to the $k$-th nearest neighbor in ring datasets with varying density. The average is over all points in each dataset. One standard deviation is given. The larger the number of samples, the smaller the distance to the $k$-th nearest neighbor.

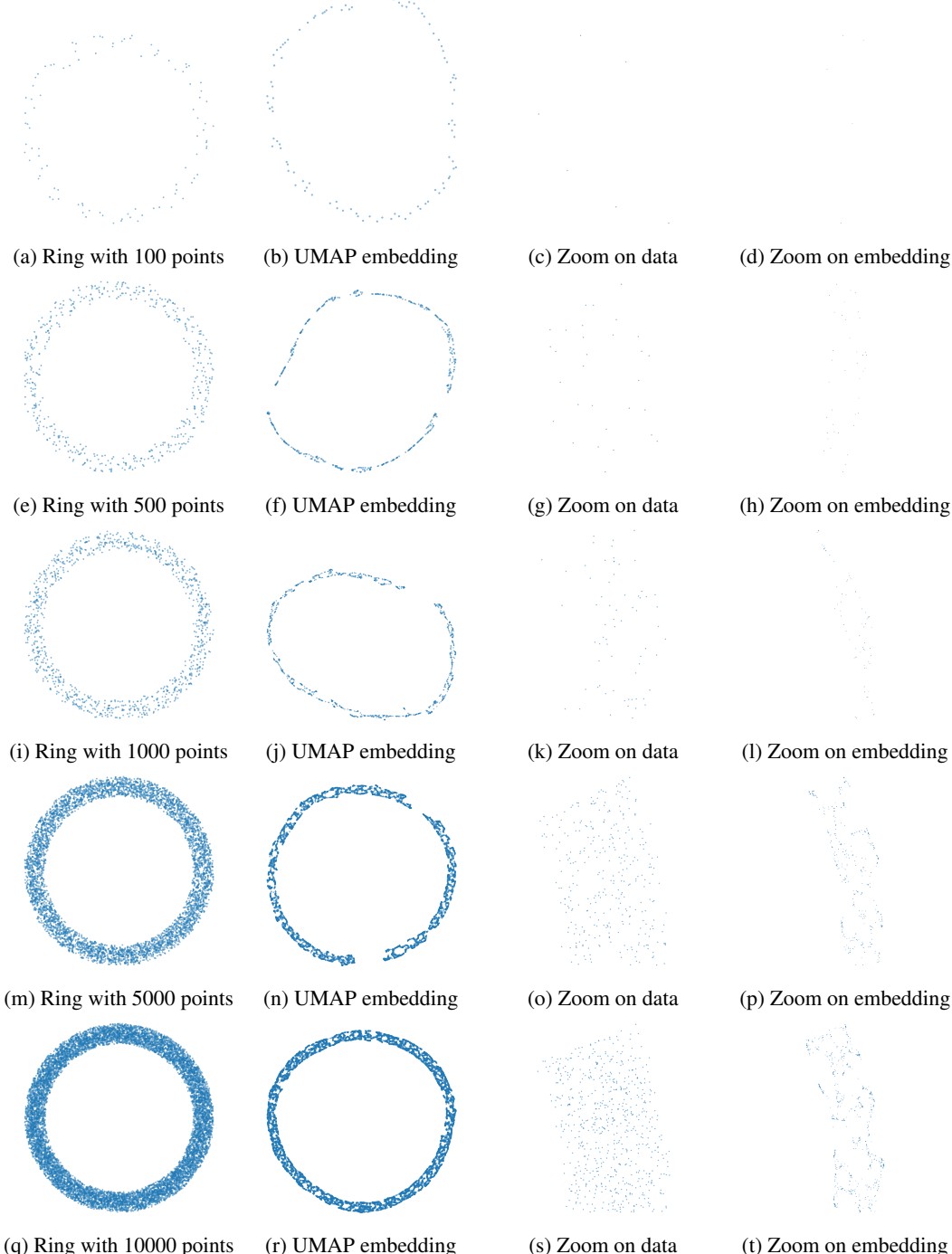

(a) Ring with 100 points    (b) UMAP embedding    (c) Zoom on data    (d) Zoom on embedding

(e) Ring with 500 points    (f) UMAP embedding    (g) Zoom on data    (h) Zoom on embedding

(i) Ring with 1000 points    (j) UMAP embedding    (k) Zoom on data    (l) Zoom on embedding

(m) Ring with 5000 points    (n) UMAP embedding    (o) Zoom on data    (p) Zoom on embedding

(q) Ring with 10000 points    (r) UMAP embedding    (s) Zoom on data    (t) Zoom on embedding

Figure 14: UMAP embedding of rings with varying number of points. **First column:** Datasets of rings in 2D. **Second column:** UMAP embedding of the dataset on the left. The embedding of the dataset with 500 points, 14f, seems most crisp, although it is not the largest dataset. **Third column:** Zoom of the original data within the angular segment $[0, \pi/8]$. **Fourth column:** Zoom on the embedding of the points that are in the third column. We see that for the larger datasets, there is over-contraction within the width of the ring in the form of clustered and line-like substructures. Figure best viewed digitally.

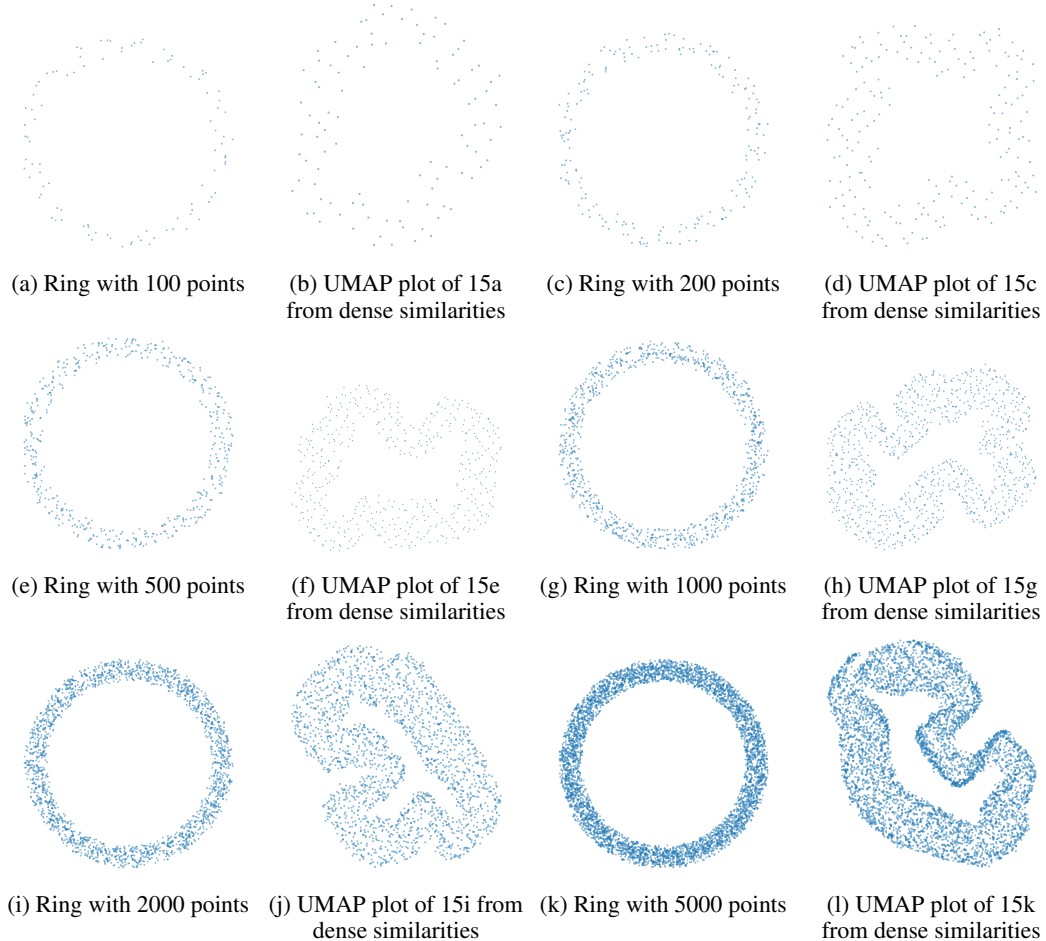

(a) Ring with 100 points

(b) UMAP plot of 15a from dense similarities

(c) Ring with 200 points

(d) UMAP plot of 15c from dense similarities

(e) Ring with 500 points

(f) UMAP plot of 15e from dense similarities

(g) Ring with 1000 points

(h) UMAP plot of 15g from dense similarities

(i) Ring with 2000 points

(j) UMAP plot of 15i from dense similarities

(k) Ring with 5000 points

(l) UMAP plot of 15k from dense similarities

Figure 15: Datasets of 2D rings with a varying number of points and their UMAP's embeddings from dense input similarities. Qualitatively the UMAP embedding does not change with the size of the dataset.

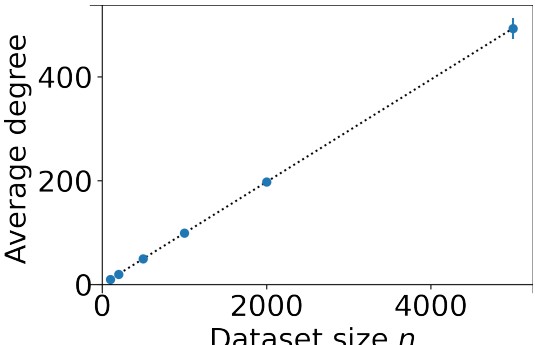

Figure 16: Average degree of a point in the 2D ring datasets of varying size of Figure 15. The average is over all points in the dataset and the error bar is one standard deviation. The dense input similarities were used. The dotted line is the best linear fit, which explains the data nearly perfectly.

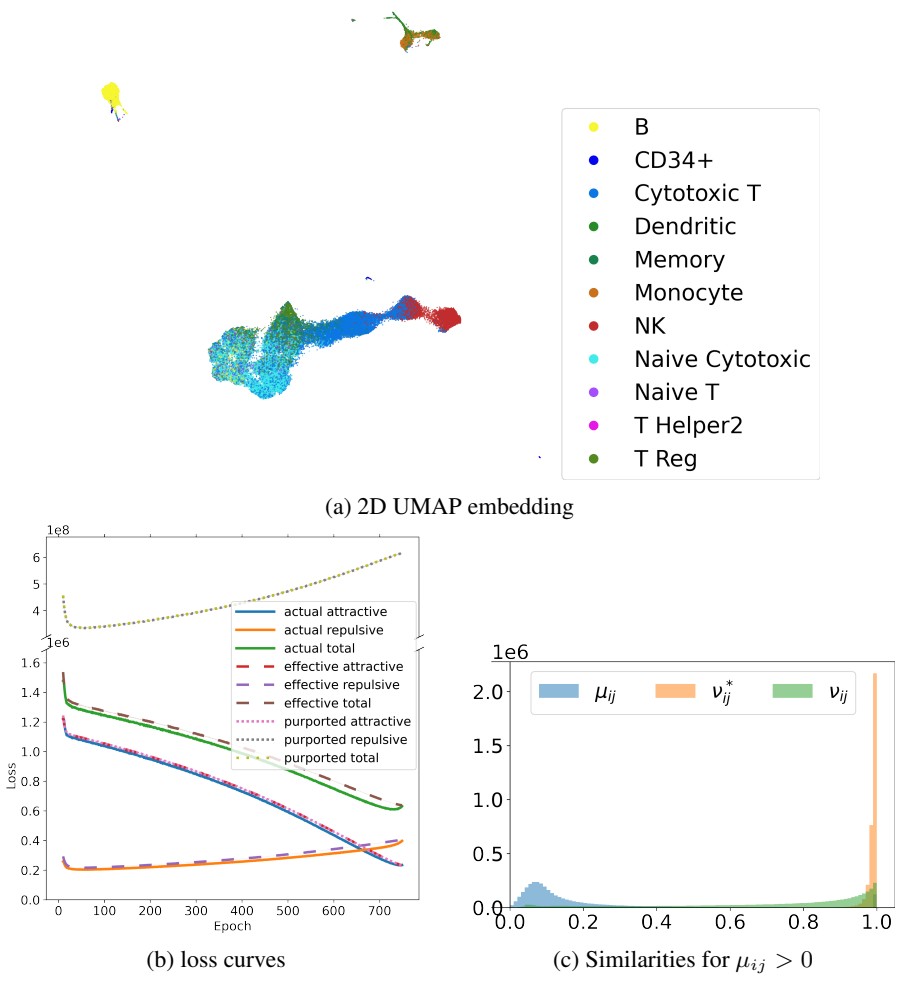

(a) 2D UMAP embedding

(b) loss curves

(c) Similarities for $\mu_{ij} > 0$

Figure 17: UMAP on the PBMC dataset [22]. **17a:** The 2D UMAP embedding exhibits over-contraction for instance in the Dendritic and the B cells. **17b:** Loss curves for the optimization leading to Figure 17a. Similar to Figure 3 our effective loss (12) closely matches the actual loss (11), while the purported UMAP loss (6) is two orders of magnitude higher. The total overlays the repulsive purported loss. An average over 7 runs is plotted with shaded areas of one standard deviation. **17c:** Histogram of high-dimensional ($\mu_{ij}$), target ($\nu_{ij}^*$) and low-dimensional ($\nu_{ij}$) similarities of the PBMC dataset for pairs with positive high-dimensional similarity. The similarities of UMAP's low-dimensional embedding reproduce the target similarities instead of the high-dimensional ones. The target similarities are heavily skewed towards one. Figure best viewed in color.

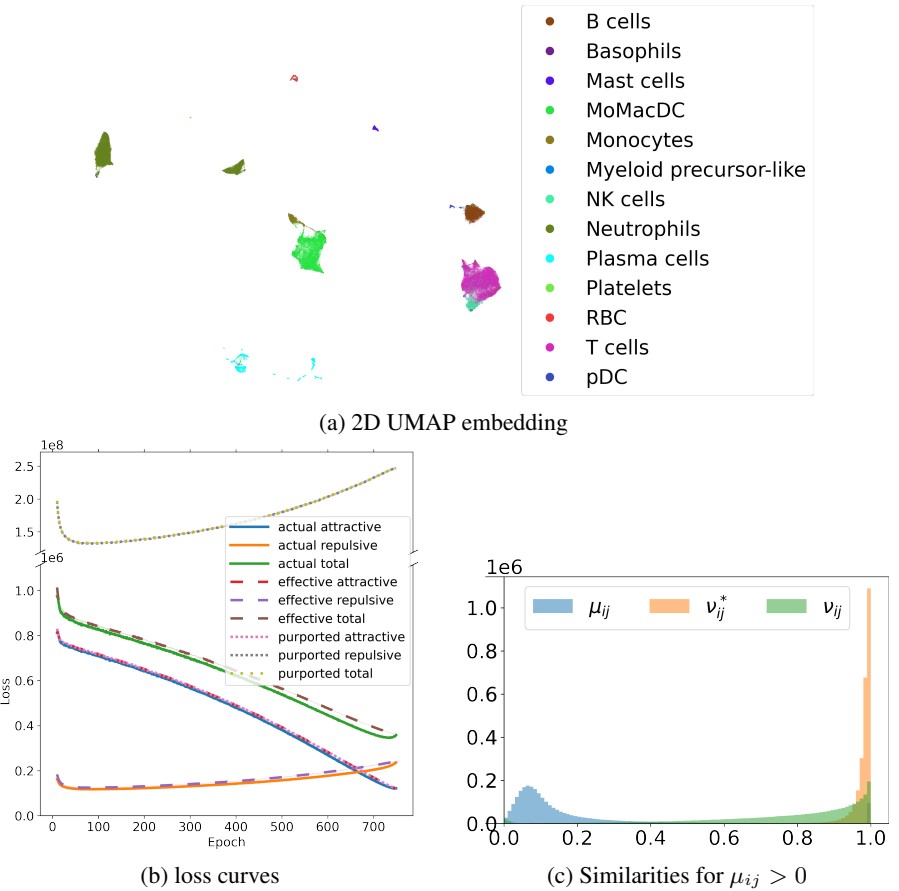

(a) 2D UMAP embedding

(b) loss curves

(c) Similarities for $\mu_{ij} > 0$

Figure 18: UMAP on the lung cancer dataset [23]. **18a:** The 2D UMAP embedding exhibits over-contraction for instance in the plasma cells and the neutophils cells near the MoMacDC cells. **18b:** Loss curves for the optimization leading to Figure 18a. Similar to Figure 3 our effective loss (12) closely matches the actual loss (11), while the purported UMAP loss (6) is two orders of magnitude higher. The total overlays the repulsive purported loss. An average over 7 runs is plotted with shaded areas of one standard deviation. **18c:** Histogram of high-dimensional ($\mu_{ij}$), target ($\nu_{ij}^*$) and low-dimensional ($\nu_{ij}$) similarities of the lung cancer dataset for pairs with positive high-dimensional similarity. The similarities of UMAP's low-dimensional embedding reproduce the target similarities instead of the high-dimensional ones. The target similarities are heavily skewed towards one. Figure best viewed in color.

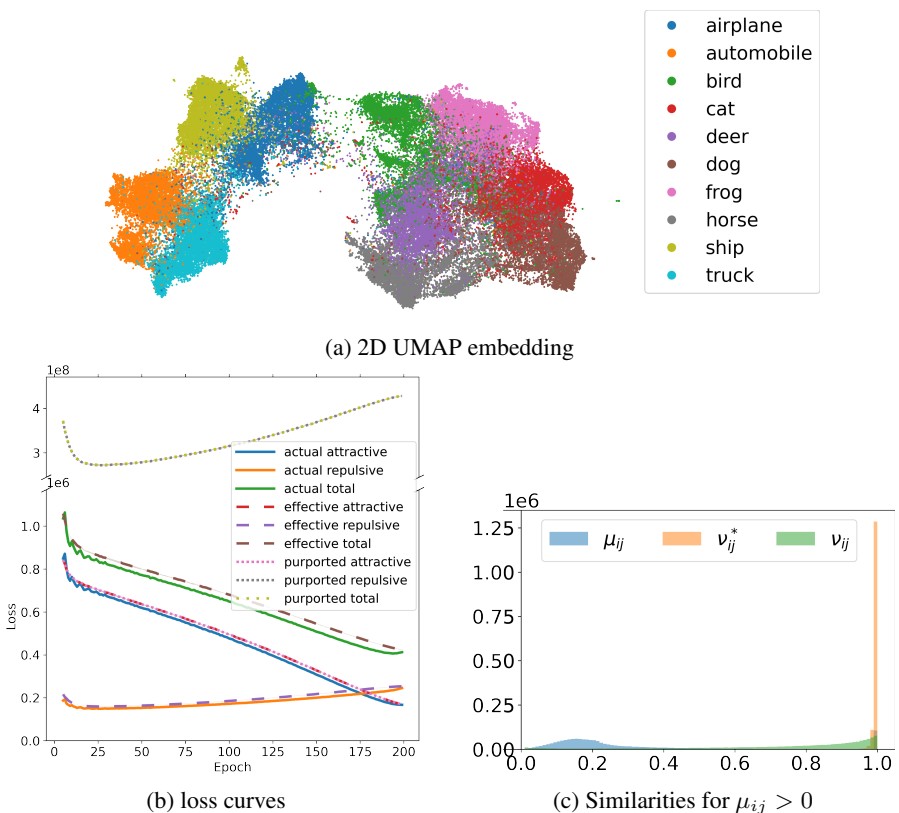

(a) 2D UMAP embedding

(b) loss curves

(c) Similarities for $\mu_{ij} > 0$

Figure 19: UMAP on Resnet50 features of CIFAR-10. **19a:** The 2D UMAP embedding exhibits descent class separation although the labels were only used to color the 2D plot. The pretrained Resnet50 extracted semantically meaningful features that UMAP was able to embed well. We see a clear separation between the vehicle and animal classes, bordered by the bird and airplane classes. **19b:** Loss curves for the optimization leading to Figure 19a. Similar to Figure 3 our effective loss (12) closely matches the actual loss (11), while the purported UMAP loss (6) is two orders of magnitude higher. The total overlays the repulsive purported loss. An average over 7 runs is plotted with shaded areas of one standard deviation. **19c:** Histogram of high-dimensional ($\mu_{ij}$), target ($\nu_{ij}^*$) and low-dimensional ($\nu_{ij}$) similarities of the CIFAR-10 dataset for pairs with positive high-dimensional similarity. The similarities of UMAP's low-dimensional embedding reproduce the target similarities instead of the high-dimensional ones. The target similarities are heavily skewed towards one. Figure best viewed in color.

## J  Additional figures

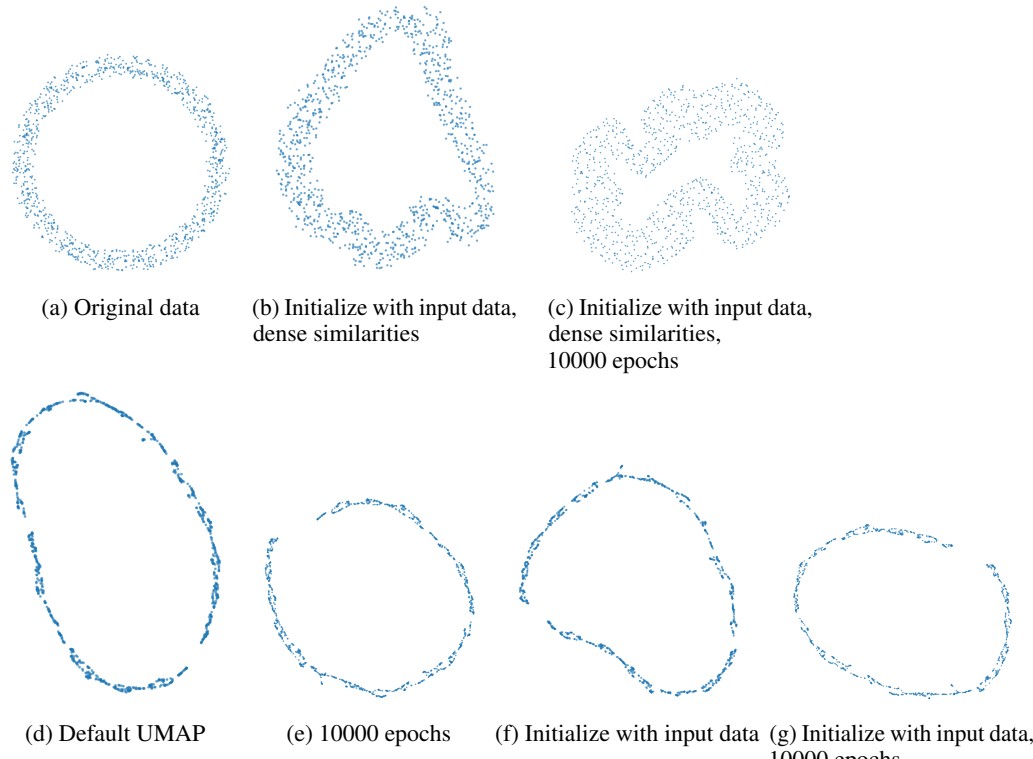

(a) Original data

(b) Initialize with input data, dense similarities

(c) Initialize with input data, dense similarities, 10000 epochs

(d) Default UMAP

(e) 10000 epochs

(f) Initialize with input data

(g) Initialize with input data, 10000 epochs

Figure 20: UMAP does not preserve the data even when embedding to the input dimension. Extension of Figure 1. **20a:** Original data: 1000 uniform samples from a ring in 2D. **20b:** Result of UMAP when initialized with the original data and using dense input space similarities computed from the original data with $\phi$. **20c:** Same as 20b but optimized for 10000 epochs. **20d:** UMAP visualization with default hyperparameters. **20e:** Same as 20d but optimized for 10000 epochs. **20f:** UMAP visualization initialized with the original data. **20g:** Same as 20f but optimized for 10000 epochs.

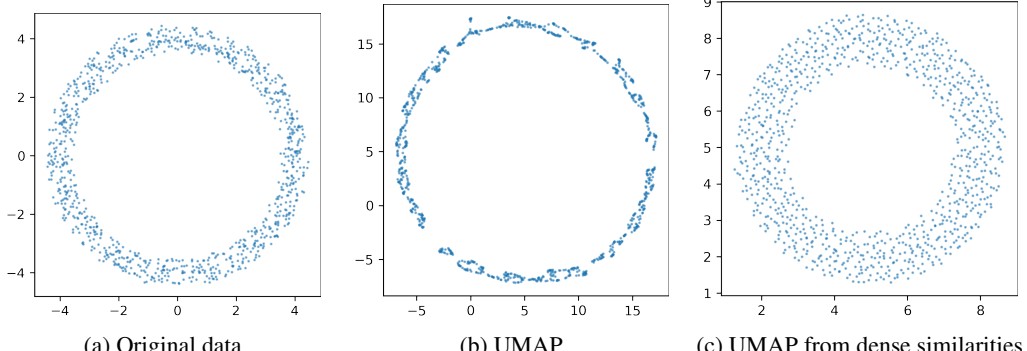

(a) Original data        (b) UMAP        (c) UMAP from dense similarities

Figure 21: Same as Figure 1 but here the tail of a negative sample is repelled from its head. **21a:** Original data. **21b:** The UMAP result looks similarly over-contracted but slightly rounder than 1b. **21c:** When initialized with the dense input similarities, the UMAP embedding has a wider than expected ring width similar to 1d but without the spurious curves. Instead the radius of the ring is smaller than in the original. Both the larger ring width and the smaller radius match the analysis in Section 6.1.

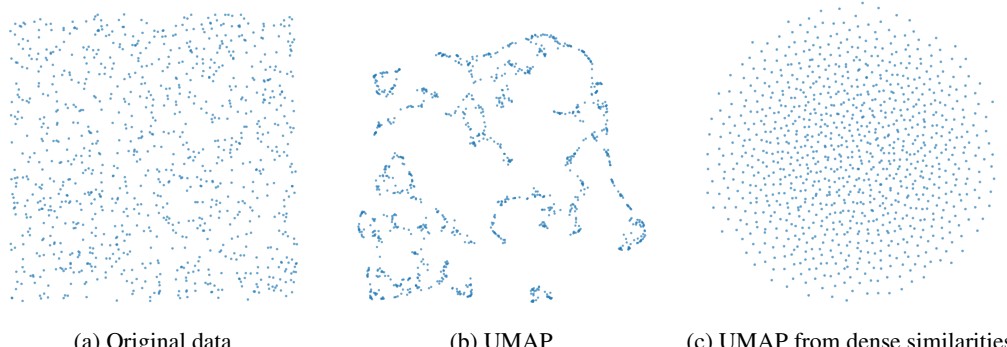

(a) Original data        (b) UMAP        (c) UMAP from dense similarities

Figure 22: UMAP does not preserve the data even when no dimension reduction is required. **22a:** Original data consisting of 1000 uniform samples from a unit square in 2D. **22b:** Result of UMAP after 10000 epochs, initialized with the original data. The embedding is much more clustered than the original data and in many locations nearly one-dimensional. **22c:** Result of UMAP after 10000 epochs for dense input space similarities computed from the original data with $\phi$, initialized with the original embedding. Reproducing the input would be optimal for the purported UMAP loss in this setting. Instead the output is circular with slightly higher density in the middle. It appears even more regular than the original data.

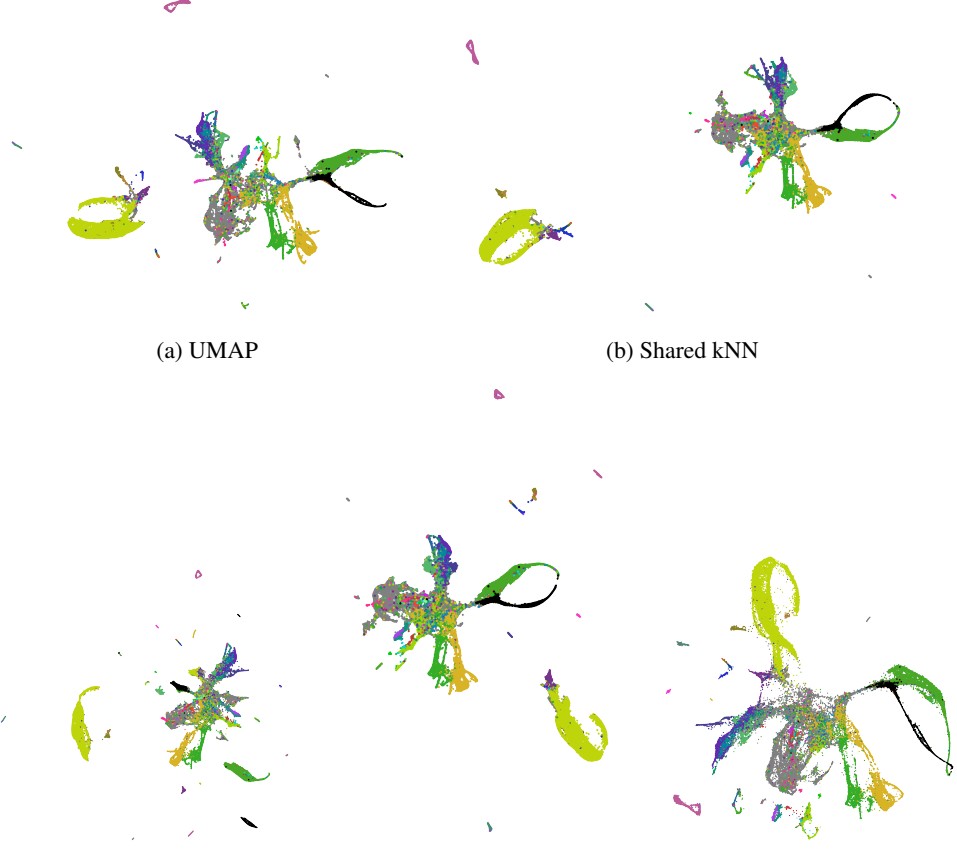

(a) UMAP                          (b) Shared kNN

(c) Permuted          (d) Uniformly random          (e) Inverted

- ABarpaaa_lineage
- Arcade_cell
- Body_wall_muscle
- Ciliated_amphid_neuron
- Ciliated_non_amphid_neuron
- Coelomocyte
- Excretory_cell
- Excretory_cell_parent
- Excretory_duct_and_pore
- Excretory_gland
- G2_and_W_blasts
- GLR
- Germline

- Glia
- Hypodermis
- Intestinal_and_rectal_muscle
- Intestine
- M_cell
- Parent_of_exc_duct_pore_DB_1_3
- Parent_of_exc_gland_AVK
- Parent_of_hyp1V_and_ant_arc_V
- Pharyngeal_gland
- Pharyngeal_intestinal_valve
- Pharyngeal_marginal_cell
- Pharyngeal_muscle

- Pharyngeal_neuron
- Rectal_cell
- Rectal_gland
- Seam_cell
- T
- XXX
- Z1_Z4
- hmc
- hmc_and_homolog
- hmc_homolog
- hyp1V_and_ant_arc_V
- nan

Figure 14: The precise value of the positive $\mu_{ij}$'s matters little: UMAP produces qualitatively similar results even for severely perturbed $\mu_{ij}$. The panels depict UMAP visualizations of the C.elegans dataset but with disturbed positive high-dimensional similarities. **23a:** Usual UMAP $\mu_{ij}$'s. **23b:** Positive $\mu_{ij}$ all set to one, so that the weights encode the shared $k$NN graph as done in [4]. **23c:** Positive $\mu_{ij}$ randomly permuted. **23d:** Positive $\mu_{ij}$ overwritten by uniform random samples from $[0, 1]$. **23e:** Positive $\mu_{ij}$ filtered as in UMAP's optimization procedure (set all weights to zero below $\max \mu_{ij}/\text{n\_epochs}$) and inverted at the minimal positive value $\mu_{ij} = \min_{ab} \mu_{ab}/\mu_{ij}$. Amazingly, the visualizations still show the main structures identified by the unimpaired UMAP. While 23c tears up the seam cells, 23e even places the outliers conveniently compactly around the main structure. The level of visual similarity for different perturbations seems particularly high when compared to Figure 9 which shows that the global placement of subgroups as well as whether loops are open or closed (e.g. seam cells and hypodermis cells) depend even on the random seed. We used random seed 0 in this figure. All C. elegans UMAP embeddings were subjectively flipped and rotated by multiples of $\pi/2$ to ease a visual comparison.

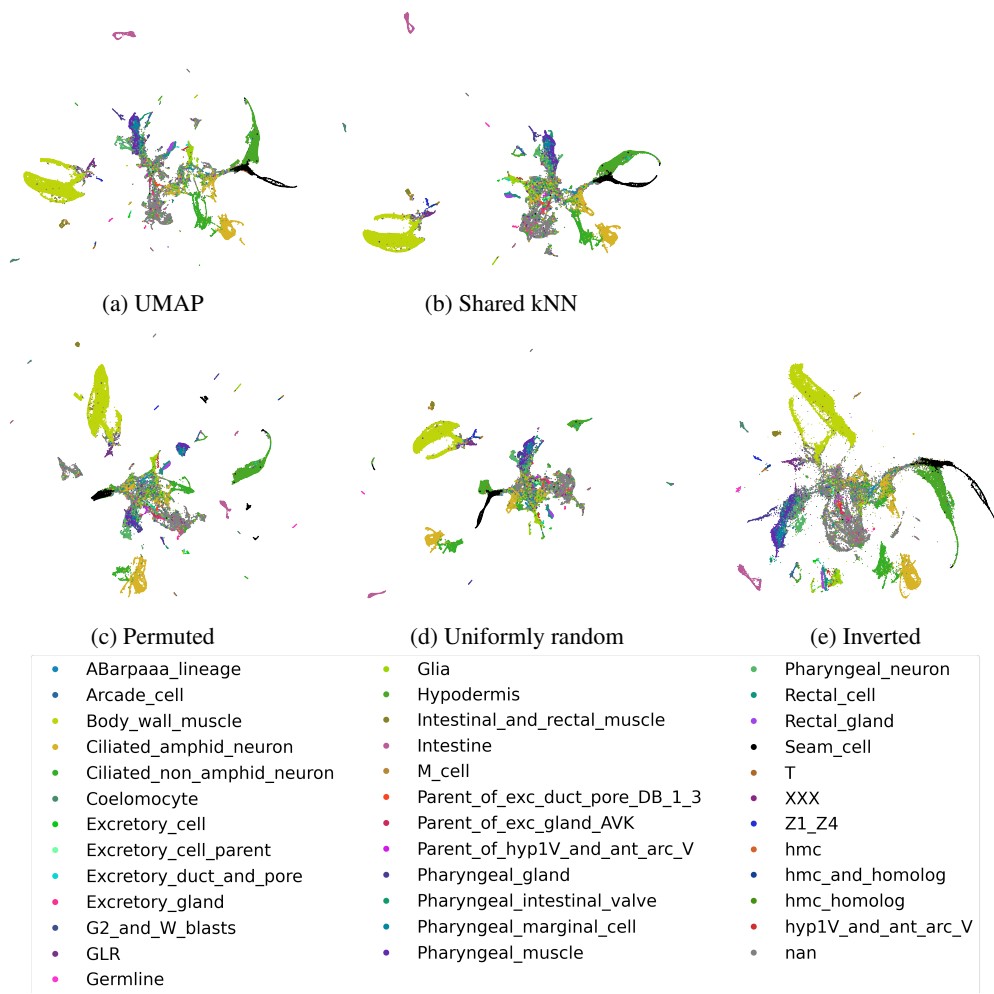

(a) UMAP       (b) Shared kNN

(c) Permuted       (d) Uniformly random       (e) Inverted

- ABarpaaa_lineage
- Arcade_cell
- Body_wall_muscle
- Ciliated_amphid_neuron
- Ciliated_non_amphid_neuron
- Coelomocyte
- Excretory_cell
- Excretory_cell_parent
- Excretory_duct_and_pore
- Excretory_gland
- G2_and_W_blasts
- GLR
- Germline

- Glia
- Hypodermis
- Intestinal_and_rectal_muscle
- Intestine
- M_cell
- Parent_of_exc_duct_pore_DB_1_3
- Parent_of_exc_gland_AVK
- Parent_of_hyp1V_and_ant_arc_V
- Pharyngeal_gland
- Pharyngeal_intestinal_valve
- Pharyngeal_marginal_cell
- Pharyngeal_muscle

- Pharyngeal_neuron
- Rectal_cell
- Rectal_gland
- Seam_cell
- T
- XXX
- Z1_Z4
- hmc
- hmc_and_homolog
- hmc_homolog
- hyp1V_and_ant_arc_V
- nan

Figure 15: Same as Figure 14 but here the tail of a negative sample is repelled from its head. There is little qualitative difference between Figure 14 and this figure overall.

# K   Societal impact

Together with $t$-SNE, UMAP is state-of-the-art for visualizing high-dimensional datasets. It is particularly popular for gene expression data. Our work contributes to the deeper understanding of UMAP, which can benefit many applications, especially in biology. We hope that our contribution will both further theoretical research in non-linear dimension reduction techniques as well as help practitioners interpret their UMAP results more faithfully.

Nevertheless, we neither provide a holistic explanation for UMAP's behavior nor establish faithfulness guarantees. Hence, we caution against overconfidence in UMAP visualizations: Insights gained from exploratory data analysis with UMAP should never be taken at face value but experimentally validated.

Exploratory data analysis is a general tool and the societal impact depends on the analyst's intention and the analyzed data. For instance, on data containing personally identifiable information, UMAP insights might constitute privacy violations.

Logging UMAP's loss as done in this work increases the computational footprint noticeably. While instructive for validating our theoretical results, we believe that in a typical use case UMAP losses do not need to be logged. We therefore recommend to avoid the additional compute unless a fine-grained analysis of UMAP's optimization procedure is needed, for instance to investigate unexpected results.