# OpenReview forum: "On UMAP's True Loss Function"
_NeurIPS.cc/2021/Conference — NeurIPS 2021 Poster_

### Official Review · Reviewer_akpc · 2021-07-10

**Rating:** 8
**Confidence:** 3

**Summary:**

UMAP is a very popular algorithm for dimensionality reduction and data visualisation. This paper analyzes the the sampling based optimisation of UMAP and show that the loss function UMAP implicitly optimises is different from the loss function it tries to optimise (because of the stochastic optimisation algorithm used). The authors derive the qualitative differences between the two and using this show that what UMAP tries to do is approximate binarised similarities of those obtained from the high dimensional setting (similarity 1 if the high dimensional similarity computed is non-zero and 0 otherwise).

**Limitations And Societal Impact:**

One complaint I have to this paper is that the writing is not self contained in a few places. For example, the authors dont define what the uniformity assumption of UMAP is and use it in the first paragraph of the introduction.

Another complaint I have is that the authors are parsimonious in using of brackets in equations like (eq 7 where a bracket around the sum over j would make it much easily parsed.

**Main Review:**

UMAP is a very popular technique for dimensionality reduction and data visualisation. As the authors point out, UMAP is derived under the assumption that the data is uniformly distributed on a low dimensional manifold in high dimensional space - which is untrue for a lot of datasets UMAP is known to perform well on. Sparse single cell RNA-seq data sets are a good example of such a dataset. Hence understanding what exactly the algorithm does is an important question - which this paper addresses.

The meat of the paper is in the authors showing that the stochastic optimisation algorithm used to optimise UMAP does not optimise the loss function it purportedly optimises. The Loss function of UMAP for a pair $i,j$ is given by

$$ \mu_{i,j} \log v_{i,j} + (1-\mu_{i,j})\log (1- v_{i,j})$$

where $\mu_{i,j}$ is the high dimensional similarity and $v_{i,j}$ are the similarity of the low dimensional embedding; $ \log v_{i,j}$ being the attraction term and $ \log(1- v_{i,j})$ being the repulsion term. Due to the computation of $\mu_{i,j}$ (based on a k-NN in high dimensions), most of them are $0$. However, that means the pre-factor multiplying the rejection term is $1$ for most of the cases.

However on analysis of the stochastic optimisation used to optimise this loss one finds that pre-factor multiplying the repulsion term is much smaller. The authors derive this in closed form. They then use it come up with interpretations of what UMAP is optimising in reality - a binarised similarity score derived in high dimensions.

I thought this was a neat analysis. The fact that it was not particularly hard mathematically just added to its beauty.

**Time Spent Reviewing:**

3.5

---

> ### Author Response · Authors · 2021-08-10
> **Response to reviewer akpc**
>
> Dear reviewer akpc,
> we thank you for the time invested and the positive review containing a concise summary of the main points of the paper.
>
> **Self-containedness of the paper:**
> We are happy to explain UMAP's uniformity assumption in more detail to make the exposition more self-contained. We envisage adding a passage like
>
> "UMAP assumes that the data lies on a low-dimensional Riemannian manifold $(R,g)$ in ambient high-dimensional space. Moreover, the data is assumed to be uniformly distributed with respect to the metric tensor $g$, or put simply, with respect to the distance along the manifold. The metric tensor $g$ in turn is assumed to be locally constant."
>
> in Section 3 and a cross-reference to this passage in the first paragraph.
>
> **Use of brackets in formulae:**
> Thank you for bringing the sparseness of brackets in the formulae to our attention. We will use brackets more generously in the revision to ease formulae parsing.

---

### Official Review · Reviewer_kpki · 2021-07-14

**Rating:** 7
**Confidence:** 4

**Summary:**

The paper investigated the UMAP's sampling-based optimization. The paper derived the effective objective function and found it to be different from the one proposed in the original paper. The difference is due to using SGD with non-uniformly-sampled positive and negative data points. The paper further derived the actual similarities which UMAP is trying to approximate. Those findings have several important implications: 1) UMAP does not try to reproduce the similarities in the original high dimensional space, 2) instead, it tries to approximate the similarities captured by the KNN graph, 3) points (1) and (2) are more so for large n (# of data points), small m (# of negative samples), and small k (# of neighbors on KNN graph).

--------------------------------
I thank the authors for providing a careful and thorough response to my review. The response has addressed most of my questions and led to new interesting ideas.


**Limitations And Societal Impact:**

The authors have adequately addressed the limitations and potential negative societal impact of their work.

**Main Review:**

The paper studies an important problem because UMAP is one of the most widely used data visualization algorithms and it is important to understand what it is trying to optimize. The theoretical analysis and empirical results are solid, and the interpretations derived from them provide a reasonable explanation on the behavior of UMAP. Overall, I think the paper is technically sound, well-written. The conclusions are useful and well-supported by both the theoretical and empirical results.

Major:

- The paper hasn't focused on how the behavior of UMAP changes with the data size n, which I think is an important aspect. Eq. (15) implies that UMAP can more faithfully recover the similarities in the high-dimensional data when the data size n is small. I think this is consistent with empirical observation of applying UMAP on real-world single-cell data. It would be nice to have a discussion on this. Specifically, at what values of (n,m,k) are the UMAP embedding close to the high-dimensional similarities, and at what values of (n,m,k) do the UMAP embedding only capture the binary KNN similarities?

- According to Eq. (15), when we further increase n, Fig. 1c will look totally random where the distances between any two points are the same. Is this what the authors observed empirically?

- According to Eq. (15), the contraction phenomenon will be more pronounced when the data size n is larger. Is this what the authors observed empirically?

- Given the different behavior of UMAP for different data sizes, it is desirable to have a version of UMAP whose behavior does not change with the data size n, either by adaptively choosing (m,k) for different n or by slightly modifying the objective function. I think the impact of the paper may be substantially improved by providing such a guideline/procedure.

- Fig. 4b: I am curious how v_ij's look in this experiment, both for the original UMAP and the one with inverted \mu_ij.



Minor:

- Line 168: "line 5" --> "Alg. 1 line 5" for clarity and similar for others.

- Equations: add parentheses to clarify which terms are inside summation.

- Supp. Figures: Please reference Supplementary Figures by "Supp. Fig." instead of just "Figure".


**Time Spent Reviewing:**

5

---

> ### Author Response · Authors · 2021-08-10
> **Response to reviewer kpki (part 1/2)**
>
> Thank you very much for the time you invested in your detailed and positive review and many valuable questions and suggestions. In the following, we will provide some responses to your comments.
>
> **Dependence on dataset size:**
> Your questions regarding the effect of the dataset size are intriguing and complement the exposition of the paper nicely, indeed.
>
> *Discussion of the choice of (n, m, k):*
> As discussed in Section 3 and Appendix B, the degree $d_i = \sum_{j=1}^n \mu_{ij}$ is approximately  $\log_2(k)$. Plugging this into Eq. (15) shows that the target similarities of UMAP's implementation are roughly
> $$ \nu^*_{ij} = \frac{\mu_{ij}}{ \mu_{ij} + \log_2(k) m / n}.$$
> This shows how the relation between target and high-dimensional similarities depends on $k$, $m$ and $n$. Note that there are no parameter settings for $m$ and $k$ that allow the (near) exact reconstruction of the whole value range of the $\mu_{ij}$'s for a fixed $n$ as this would require
> $$\log_2(k) m / n = 1- \mu_{ij} \forall i,j.$$
> For instance, we can see in Fig. 4 (a) that most non-zero high-dimensional similarities are either $1$ or close to $0.05$ for the C. elegans dataset with $k=30$ and $n = 86024$. To preserve the $\mu_{ij}$'s that equal $1$, we would need
> $$m = (1-1) * 86024 / \log_2(30) = 0,$$
>  while to preserve the $\mu_{ij}$'s near $0.05$, we would need
> $$m = (1- 0.05) * 86024 / \log_2(30) \approx 16500$$
>  negative samples, which in addition would be prohibitively slow computationally. Changing $k$ would also change the algorithm's notion of locality, which has implications on both speed and appearance.
>
> As another example, consider UMAP's default setting of $k=15$ and $m=5$. Then already for $n=500$ each $\mu_{ij} > 0.2$ would be mapped to a $\nu_{ij}^* > 0.83$. So even for relatively small datasets the over-contraction is strong. This is particularly significant, as the UMAP authors caution against the use of UMAP on datasets smaller than $500$ points due to various approximations in the algorithm at the end of Section 6 in [11].
>
> *Increasing $n$ in Fig 1.(c):*
> For this figure, we used the dense input similarities $\mu_{ij} = \phi(|| x_i - x_j||)$ instead of the typical sparse ones based on the $k$NN graph (confer also the different analysis for this figure in Section 6.1). Therefore, the degree $d_i$ for a point $i$ scales with the dataset size and the effective repulsive pre-factor $(d_i + d_j) *m / (2n)$ does not decrease with the dataset size but remains approximately constant. Hence, we do not expect any qualitative difference in Fig 1.(c) as we vary $n$. Indeed, experiments in which we changed the number of points of the ring from $100$ to $5000$ confirm this prediction and the average degrees indeed scale with $n$.
>
> *Increasing $n$ in normal UMAP visualizations:*
> To study the behavior of UMAP as the dataset size varies, we conducted two sets of experiments.
> In the first, we up- or down-sampled the number of data points in the toy ring and the C. elegans dataset.
> As predicted by Eq. (15), the histograms of the target similarities were increasingly skewed towards one as the dataset size increased. The low-dimensional similarities of the optimized embedding followed suit: The larger the dataset, the higher the share of $\nu_{ij}$'s close to $1$. This supports our theoretical predictions and in particular your observation that over-contraction is more severe for large datasets.
> Visually, however, the results were more subtle. On the more complex C.elegans dataset, we did not see any obvious signs of over-contraction for subsamples of size smaller than $1000$ cells. But on such small subsets the overall structure of the dataset was also barely apparent. From $1000$ cells onward, we saw several locally one-dimensional areas indicating over-contraction. However, it was not the case that all structures got "slimmer" as the subsample size increased. On the toy ring dataset, we observed a similar trend: For sizes up to $300$ points the ring width of the embedding tended to decrease and from $800$ points onward it increased again until it reached about $2/3$ of the original ring width for a ring with $10000$ points.
>
> Puzzled by the apparent mismatch between the similarity histograms, which showed ever increasing over-contraction, and the visualizations themselves, which looked like the over-contraction reduces for large datasets, we zoomed into the embeddings for rings with many points. It turned out that the embedding points were clumped together in tight clusters or formed lines within the width of the ring. The zoomed-in plots look very similar to Supp. Fig. 13 b), which depicts the UMAP result of a dataset sampled uniformly from a square in two dimensions. Our conclusion is that the over-contraction does increase with the dataset size, but its effects might look different on a global scale. Sometimes one can see locally one-dimensional parts or tight clusters on the global scale and sometimes they only become visible when zooming in.
>
> We conjecture that the reason for whether one observes the locally one-dimensional regions on a global scale has to do with the relative size of the local neighborhood, $k$, and the size of the slim structure. If the local neighborhood is large relative to the width of the slim structure, it will get contracted to a line. If it is small, the over-contraction happens within the slim structure.
> Another way of looking at this phenomenon is that for fixed $k$ but increasing sample size of the toy ring, the distance to the $k$-nearest neighbor and thus the radius of the local neighborhood decrease relative to the fixed width of the whole ring. We computed the mean distance to the $k$-th nearest neighbor over all data points. At $300$ data points, when the width of the embedding has essentially reached $0$, the mean $k$-th nearest neighbor radius is about $0.75$ and thus nearly as wide as the original ring width of $1$. At $800$ points, when the embedding ring width is increasing again, the average $k$-th nearest neighbor radius is only $0.4$, much smaller than the original ring width of $1$. We note for completeness that for some toy rings with few points, the ring structure was torn up to a line or even a set of clusters.
> Choosing an appropriate $k$ might help to overcome visible over-contraction by moving it to a scale much smaller than the global scale. But since a dataset might have subregions of different density or scale, choosing a fixed $k$ for the whole dataset is difficult.
>
> In the second experiment, we tried to decouple the effect of over-contraction from the scale of locality. We created two-dimensional datasets of multiple rings with $1000$ points per ring as in Fig. 1a) and separated the rings sufficiently, so that no point has any of its $k$ nearest neighbors in a different ring. Again, we observe that as the number of rings and thus the dataset size increase, the distributions of the target and low-dimensional similarities get skewed more towards $1$. As the radius of the local neighborhoods does not decrease with the dataset size, we now clearly see that the larger the dataset the more the embedding of individual rings look like a circle with zero width.

---

> > ### Author Response · Authors · 2021-08-10
> > **Response to reviewer kpki (part 2/2)**
> >
> > *UMAP version without dependence on the dataset size:*
> > First, we would like to point out that while there certainly is some information in the exact values of the $\mu_{ij}$, optimizing UMAP's purported loss function, if it were feasible, would not be desirable.
> > Consider the toy experiment in Table 1. The diverged embedding has much lower loss according to the purported loss function than the actual UMAP embedding, so that the low-dimensional similarities of the diverged embedding are closer to the fancy high-dimensional similarities than those of the actual UMAP embedding. However, a diverged embedding is useless while the actual UMAP embedding is usually insightful. This aligns with the experiment of [3], in which a Barnes-Hut approximation to the purported UMAP loss function led to a diverged and thus pointless embedding. We claim that UMAP produces good results not despite but only because its implementation optimizes a different loss function than intended.
> >
> > Adapting $m$ and $k$ to the dataset size $n$, e.g. by requiring $\log_2(k) m / n$ to be a constant, might be a way to make UMAP's behavior less dependent on the dataset size. But as mentioned above, increasing $m$ and $k$ increases the run-time of the algorithm and in the case of $k$ would also change the notion of locality. This might be desirable in cases where we sample a fixed dataset more or less densely like when we subsampled the C.elegans dataset above. But it might also be undesirable if the dataset consists of essentially independent subsets, like in the experiment with multiple rings above. Moreover, the value $\log_2(k) m / n$ would become a hyperparameter that controls which range of high-dimensional similarities is decreased, kept constant or increased, as discussed above.
> > The most severe problem that we see with keeping the repulsion pre-factor $(d_i + d_j) m / (2n)$ constant is that this means that the total amount of repulsion acting on a single embedding point scales with the dataset size while the total amount of attraction remains constant. This way, we risk to arrive at diverged embeddings for sufficiently large datasets. While the inverse relation of the repulsion pre-factor with the dataset size can lead to over-contraction on the $k$NN graph edges, it also counter acts the quadratic number of non-$k$NN graph edges.
> >
> > Your and reviewer 8grW's inquiries about improving UMAP sparked an idea to change the optimization procedure of UMAP by decreasing the repulsion per edge only on the non-$k$NN graph edges but not on the $k$NN graph edges. Concretely, we envisage to sample positive edges as usual and obtain negative samples again as before for each positive sampled edge. In addition, we suggest to explicitly sample the $k$NN graph edges for repulsion with probability $1-\mu_{ij} - (d_i + d_j)m / (2n)$. This would correct the repulsion weight for edges that are part of the $k$NN graph. The loss function would read
> > $$- 2 \sum_{ij \in k\text{NN graph}}\left( \mu_{ij} \log(\nu_{ij}) + (1-\mu_{ij}) \log(1-\nu_{ij})\right) - 2\sum_{ij \notin k\text{NN graph}} \left((d_i + d_j) m / (2n) \log(1- \nu_{ij})\right).$$
> > Now the target similarities are precisely the high-dimensional similarities $\mu_{ij}$ for all pairs $i,j$. Hence, in an optimal embedding, all $\mu_{ij}$'s are exactly reproduced and not just a binary version of them. Hopefully, this will improve the embedding quality and decrease the observation of over-contraction.
> > Crucially, the amount of repulsion per data point is kept constant and is not increasing with $n$. For the quadratic number of non-$k$NN graph edges, the individual repulsion pre-factors still decrease with $1/n$. The total loss function is a sum of normalized cross-entropy terms for the edges in the $k$NN graph and of non-normalized and effectively drastically down-weighted cross-entropy terms for the edges not in the $k$NN graph.
> > Since not all $O(n^2)$ negative pairs are considered explicitly and the additional sampling of the $k$NN graph edges for repulsion only scales with $O(n)$, the complexity of computing the embedding would still scale linearly as $O(k m n * \text{num}_{\text{epochs}})$ and not quadratically in $n$. Unfortunately, we did not have the time to implement and explore this version of UMAP, but plan to do so in future work.
> >
> > We are happy to include our discussion of varying dataset size in the revision and thank you for suggesting this interesting line of investigation!
> >
> > **$v_{ij}$'s in Fig 4b):**
> > The histogram of the $v_{ij}$'s of the original UMAP is shown in Fig. 4a). That for the case of inverted $\mu_{ij}$'s it is barely distinguishable from the one in Fig. 4a), which is why we omitted both in Fig. 4b).
> >
> > **Minor comments:**
> > Thank you very much for your keen eye. We will implement your suggestions in the revision.

---

### Official Review · Reviewer_8grW · 2021-07-16

**Rating:** 6
**Confidence:** 4

**Summary:**

This paper addresses the effects of the negative sampling strategy during UMAP’s optimization procedure. The authors provide a theoretical explanation of how this procedure distorts the objective function portrayed in UMAP’s original paper. The paper derives the actual objective function being optimized when using negative sampling, showing how it corresponds to lower repulsive forces that the ones intended in the cross-entropy loss function. Such behavior causes over contracted embeddings, which do not consider connectivity beyond the Knn neighbors.

**Limitations And Societal Impact:**

Yes

**Main Review:**

The paper leverages the empirical observations found in [3], giving a concise analysis for several previous claims. In particular, the paper answers various questions, such that:
-	Why does using binarized affinities not lead to noticeable changes in the embedding?
-	What is the exact repercussion of using negative sampling in the final embeddings produced by UMAP?

The paper is well-written and clear. It does provide good insight into UMAP. However, the results do feel somewhat incremental and thus limited in its significance. Are their implications of these results that go beyond UMAP, perhaps to other visualization methods? I.e., can these results be used to either improve UMAP or other visualization methods?

Other comments:

Theorem 5.1 seems out of place. At least a brief comment about parametric UMAP and the notation in the theorem is needed. For instance, there is no definition of f(x), u_{tot}.


**Time Spent Reviewing:**

4.5

---

> ### Author Response · Authors · 2021-08-10
> **Response to reviewer 8grW**
>
> Thank you very much for your positive review, the time you invested and your helpful feedback! In the following, we will provide some responses to your comments.
>
> **Further implications of our results:**
> We feel it is of high importance to understand the mechanics of a method as popular as UMAP, not just from a largely empirical observation as in [3], but also on a theoretical level. Our results guide possible improvements of UMAP away from even more sophisticated high-dimension similarities and towards the optimization scheme. Having identified the negative sampling as the culprit for the binarization of the input similarities $\mu_{ij}$, we can try to target this specifically in order to transport more information from high- to low-dimension than just the binary $k$NN graph weights.
>
> *Improved UMAP:*
> Your and reviewer kpki's inquires about improving UMAP sparked an idea to change the optimization procedure of UMAP so as to decrease the repulsion per edge only on the non-$k$NN graph edges but not on the $k$NN graph edges. Concretely, we envisage to sample positive edges as usual and obtain negative samples again as before for each positive sampled edge. In addition, we suggest to explicitly sample the $k$NN graph edges for repulsion with probability $1-\mu_{ij} - (d_i + d_j)m / (2n)$. This would correct the repulsion weight for edges that are part of the $k$NN graph. The loss function would read
> $$- 2 \sum_{ij \in k\text{NN graph}}\left( \mu_{ij} \log(\nu_{ij}) + (1-\mu_{ij}) \log(1-\nu_{ij})\right) - 2\sum_{ij \notin k\text{NN graph}} \left((d_i + d_j) m / (2n) \log(1- \nu_{ij})\right).$$
> Now the target similarities are precisely the high-dimensional similarities $\mu_{ij}$ for all pairs $i,j$. Hence, in an optimal embedding, all $\mu_{ij}$'s are exactly reproduced and not just a binary version of them. Hopefully, this will improve the embedding quality and decrease the observation of over-contraction.
> Crucially, the amount of repulsion per data point is kept constant and not increasing with $n$. For the quadratic number of non-$k$NN graph edges, the individual repulsion pre-factors still decrease with $1/n$. The total loss function is a sum of normalized cross-entropy terms for the edges in the $k$NN graph and of non-normalized and effectively drastically down-weighted cross-entropy terms for the edges not in the $k$NN graph.
> Since not all $O(n^2)$ negative pairs are considered explicitly and the additional sampling of the $k$NN graph edges for repulsion only scales with $O(n)$, the complexity of computing the embedding would still scale linearly as $O(k m n * \text{num}_{\text{epochs}})$ and not quadratically in $n$. Unfortunately, we did not have the time to implement and explore this version of UMAP, but plan to do so in future work.
>
> *LargeVis:*
> Our analysis is also applicable to the visualization method LargeVis [a], which uses a very similar optimization scheme as UMAP, but with slightly different negative sampling. Its loss function is given in [a] as
> $$- \sum_{ij \in k\text{NN graph}} \mu_{ij} \log(\nu_{ij}) - \sum_{ij \notin k\text{NN graph}}\gamma \log(1-\nu_{ij}),$$
> with $\gamma$ a constant, set to $7$ per default in [a], and, different from UMAP, $\mu_{ij}$ and $\nu_{ij}$ are the high- and low-dimensional similarities from $t$-SNE [16]. For optimization, they use negative sampling and arrive at the objective function
> $$- \sum_{ij} \mu_{ij} \left(\log(\nu_{ij}) + \sum_{l=1}^m E_{j_l \sim P(a)}( \gamma \log(1-\nu_{ij_l}))\right),$$
> where $P(a) \propto d_a^{0.75}$. As done in our paper, we can compute the expectation and arrive at the following closed form loss function for LargeVis:
> $$ - \sum_{ij} \mu_{ij} \log(\nu_{ij}) + \frac{\gamma m (d_id_j)^{0.75}(d_i^{0.25} + d_j^{0.25})}{2 \sum_{l=1}^n d_l^{0.75}} \log(1-\nu_{ij}).$$
> LargeVis does not motivate its loss function as a sum of cross-entropy losses for each edge, but instead as attraction on edges of the $k$NN graph plus constant repulsion on all non-$k$NN edges. Nevertheless, the negative sampling based optimization turns this into a sum of non-normalized cross-entropy losses as for UMAP. We strongly believe that the repulsion pre-factor is very small, so that what we called "target similarities" would look binary for LargeVis as well. This puts LargeVis in even closer proximity to UMAP as the different choice of $\mu_{ij}$ in both methods matters little. We will include these comments on LargeVis into the revised version of the manuscript. In future work, we plan to investigate why [3] found that using the binary $k$NN graph weights also does not change the visualizations much for $t$-SNE.
>
> **Placement of Theorem 5.1:**
> Thank you for alerting us to this trip in the reading flow. We will explain the context switch in more detail, state the definition of $f(x)$ and recall that of $\mu_{\text{tot}}$ in Theorem 5.1.
>
>
> **References:**
> [a] Tang, Jian, et al. "Visualizing large-scale and high-dimensional data." Proceedings of the 25th International Conference on World Wide Web. 2016

---

### Official Review · Reviewer_oNq1 · 2021-07-17

**Rating:** 6
**Confidence:** 3

**Summary:**

Authors tries to reveal the true loss function of UMAP projection given its success in data visualization and dimensionality reduction.

**Limitations And Societal Impact:**

1) In Fig.2(a), the projections of seam cells is tightly clustered but still has non-zero variance along two different directions which is not a line in 2D space. Authors stated that the embeddings are locally one-dimensional which is ambiguous.
2) In Fig.3, the curve corresponding to "purported total" is missing due to the compressed y-axis based on the range of "purported total" and "purported attractive".
3) Authors extensively experimented using the C.elegans dataset and it would be helpful to include at least one more scRNA-seq dataset as demonstration.
4) This manuscript didn’t answer the question that why balanced attraction and repulsion would lead to a “good” embedding. This is beyond the scope of this paper as stated in the discussion section. The quantitative metric related to “good” embedding authors frequently used was similarity. Authors could report similarity values for the C.elegans dataset to see if the similarity values are in concordance with visual impressions of good UMAP embeddings.


**Main Review:**

Authors demonstrated that UMAP didn't preserve the high-dimensional similarity based on a toy dataset (2D-to-2D UMAP transformation). This provides a neat example to reveal the discrepancy between theory and the empirical behavior. Authors derived the true updating function for UMAP in a close form which shed light in understanding the behavior of UMAP in depth.

**Time Spent Reviewing:**

1

---

> ### Author Response · Authors · 2021-08-10
> **Response to reviewer oNq1**
>
> Thank you very much for the time you invested, your positive review and your helpful comments. We were also particularly happy about the 2D-to-2D experiments.
>
> **1. Notion of "locally one-dimensional":**
> What we meant by "locally one-dimensional" is that in many places, for instance within the seam cells, the embedding looks as if it lies approximately on an embedded one-dimensional manifold. Indeed, there is some 2D spread among the seam cells in Fig 2.(a). But close to the tip, for instance, that spread is much smaller than the width of a single scatter point, so that we feel speaking of that part as looking one-dimensional is justified. Closer to the middle of the seam cells the spread seems wider. But upon close inspection, one sees that there are actually two dense parallel lines of points with very few points in between.
> In any case the spread in the direction of second largest variation is much smaller than one might expect from the PCA. Moreover, we observe a similar phenomenon also for parts of the ciliated (non-) amphid neurons, the glia, germline, hypodermis, pharyngeal gland as well as the "hyp1V and ant arc V" cells (see legend in Supp. Fig. 14). We will explain what we mean by locally one-dimensional in the revised version of the paper.
>
> **2. "purported total" loss curve in Fig. 3:**
> As mentioned in the caption, the "purported total" loss curve is overlaid by the "purported repulsive" loss curve, as their difference is barely visible in the compressed y-scale in the upper part of the plot. We will make this more easily visible by alternating between dots for the "purported total" and "purported repulsive" loss curves in the revision.
>
> **3. Additional scRNA-seq datasets:**
> In addition to the CIFAR10 experiment in Appendix F, we have also measured the various UMAP loss curves on  the PBMC dataset [c] and  the lung cancer dataset [d]. We used the preprocessed versions of these datasets from [12]. In both cases, the qualitative results predicted by our theory hold: the repulsion is much decreased, the high-dimensional similarities are essentially binarized, the low-dimensional similarities $\nu_{ij}$ are thus skewed towards $1$ for the edges of the $k$NN graph and parts of the embedding are visibly over-contracted. We are happy to include the results of these two additional scRNA-seq datasets in a revised appendix.
>
>
> **4. a) Why does balanced attraction and repulsion lead to "good" embeddings:**
> As we stated in the discussion and you acknowledge, this difficult question is beyond the scope of our paper. The question is particularly hard because the quality of an embedding can be highly domain specific and very subjective. We will elaborate on the main point of the discussion: preventing repulsion from scaling with the dataset size $n$. A single embedding point has attractive interaction with about $k$ other embeddings but repulsive interaction with all $n-1$ other points. If the strength of attraction for an edge in the $k$NN graph were on average of the same magnitude as the strength of repulsion for an average pair of embedding points, the sum of pre-factors for the repulsion that acts on a single embedding point is $n/k$ times larger than the sum of pre-factors of the attraction acting on it. Plausibly, in such a situation all points would diverge from each other producing a useless visualization. This intuition is underpinned by the low purported loss function value that we computed for such a diverged embedding in the upper left entry of Table 1 and by the Barnes-Hut approximation of UMAP that [3] implemented and which led to a diverged embedding. Hence, preventing the amount of repulsion to scale with the dataset size seems to be at least a necessary condition for a good embedding.
>
> **4. b) Measuring the quality of the embedding via "similarity":**
> Unfortunately, we fear that we do not quite understand your suggestion. Let us clarify our use of "similarity" in the paper: The quantities $\mu_{ij}$ and $\nu_{ij}$ measure how similar each pair of points $i,j$ is in high- and low-dimensional space, respectively. UMAP then aims to make these two notions of similarity align between high- and low-dimensional space, by minimizing their discrepancy, measured via cross-entropy loss functions for each pair of points $i, j$. In that way, the loss values for an embedding might be used to measure the quality of an embedding. But this only makes sense if one compares different low-dimensional embeddings and corresponding similarities $\nu_{ij}$ to the same high-dimensional similarities $\mu_{ij}$ via the same loss function. In particular, the fact that the actual and effective losses in Fig. 3 are much lower than the purported loss function is not an indication of a good embedding per se.
>
> There are measures that aim to quantify how faithful and thus "good" a low-dimensional visualization of a high-dimensional dataset is. For instance, in [1] the Pearson correlation between all pairwise distances in high- and low-dimensional space for a subsample of the dataset (for efficiency) is considered. In [b], the Spearman correlation is used instead and considered a "global" measure for the faithfulness of the embedding. As "local" measure [b] computes the share of $k$-nearest neighbors of a point in high dimension that are also $k$-nearest neighbors in the low-dimensional embedding. [b] use $k=10$.
> We have computed these measures for the C.elegans dataset for the two-dimensional visualizations with PCA and with UMAP. We used subsamples of size $10000$ from the $86024$ cells in the C. elegans dataset for the correlation measures and report mean and standard deviation over $5$ such subsamples. For the $k$NN accuracy we used  $k=10$ as in [b] as well as $k=30$, because on the C. elegans dataset we used $30$ neighbors in UMAP's approximate $k$NN graph computation. The results are summarized in the following table.
>
> Measure | PCA | UMAP
> --- | --- |---
> Pearson's r | $0.577 \pm 0.001$  | $0.558 \pm 0.003$
> Spearmans's r | $0.611 \pm 0.001$ | $0.551 \pm 0.003$
> kNN accuracy ($k=10$) | $0.006$ | $0.157$
> kNN accuracy ($k=30$) | $0.014$ | $0.257$
>
> As previously observed by [b], PCA is better at preserving the global structure as indicated by the higher correlation coefficients, but much worse than UMAP at preserving the local neighborhood structure. This is unsurprising as UMAP tries to reproduce local similarities in low-dimensional space, which are positive only on the $k$NN edges. Our binarization analysis shows that UMAP essentially just uses the binary high-dimensional $k$NN structure to guide its low-dimensional embedding.
> The question whether and why the $k$NN accuracy  or the correlation measures might correlate with what practitioners consider useful embeddings is again difficult to answer.
> We hope that this has addressed your concern. In case there is a misunderstanding, we would appreciate a clarification very much!
>
> **References:**
> [b] Kobak, Dmitry, and Philipp Berens. "The art of using t-SNE for single-cell transcriptomics." Nature communications 10.1 (2019): 1-14.
> [c] Zheng, G. X. et al. Massively parallel digital transcriptional profiling of single cells. Nat. Commun. 8, 14049 (2017).
> [d] Zilionis, Rapolas, et al. "Single-cell transcriptomics of human and mouse lung cancers reveals conserved myeloid populations across individuals and species." Immunity 50.5 (2019): 1317-1334.

---

> > ### Comment · Reviewer_oNq1 · 2021-08-27
> > **Resolve the confusion on "similarity"**
> >
> > What I meant was simple: you could list the similarity values (or mean similarity values) for each panel in Figure 2. In this way you can check if a “good” embedding (justified visually) corresponds to a “good” similarity value. I appreciate your extra effort to make your statement clear. Besides Pearson and Spearman correlation values, KL divergence is also one potential metric connected to t-SNE. I don’t think the conclusion would change much. Regardless of this, I think my concerns have been properly addressed.

---

> > > ### Author Response · Authors · 2021-08-28
> > > **Thank you for the clarification!**
> > >
> > > We now understand your suggestion better, thank you for clarifying! Consequently, we include three additional metrics that measure the embedding quality in the table below:
> > > 1. the purported UMAP loss,
> > > 2. the effective UMAP loss as predicted by our paper,
> > > 3. the KL divergence.
> > >
> > > For all three, we use the high-dimensional similarities of the normal UMAP setting, corresponding to Fig 2.a), as target. We also compute all measures for the embedding obtained from inverting the positive UMAP weights (Fig. 2b). Note that for the optimization of the embedding in Fig. 2b), we did not use the normal UMAP weights. But we still compute the additional three metrics with the normal UMAP weights for consistency. The full table is
> > >
> > > Measure | PCA | UMAP inv | UMAP
> > > --- | --- |--- | ---
> > > Pearson's r | $0.577 \pm 0.001$  | $0.633\pm0.003$ | $0.558 \pm 0.003$
> > > Spearmans's r | $0.611 \pm 0.001$ | $0.612 \pm 0.002$ |$0.551 \pm 0.003$
> > > kNN accuracy ($k=10$) | $0.006$ | $0.080$ | $0.157$
> > > kNN accuracy ($k=30$) | $0.014$ | $0.185$ | $0.257$
> > > purported UMAP loss | $5.37 \times 10^8$| $4.32 \times 10^8$ | $3.63 \times 10^8$
> > > effective UMAP loss | $12.4 \times 10^5$ |  $3.85 \times 10^5$ | $ 3.19 \times 10^5$
> > > KL divergence | $0.693$ | $0.575$ | $0.547$
> > >
> > > We observe surprisingly high correlation scores for the inverted UMAP setting. The kNN accuracies of the inverted UMAP setting are still much larger than those of PCA, but worse than for UMAP with the normal weights. Possibly, inverting the weights obstructed reproducing the local neighborhoods somewhat despite the effective binarization of the high-dimensional similarities. In the purported UMAP loss, the effective UMAP loss and the KL divergence the normal UMAP embedding has lowest loss, the inverted UMAP embedding higher and the PCA significantly higher loss. In the effective UMAP loss the two UMAP embeddings are particularly close while PCA is much worse. This is in accordance with the visual similarity of the embeddings and might suggest that our effective UMAP loss aligns well with what one might call a "good" embedding.

---

### Author Response · Authors · 2021-08-10
**Overall response**

We thank all reviewers cordially for the time and effort they invested in reviewing our paper and appreciate their positive reviews. We will address their concerns in individual replies. Numbered citations will refer to the references of the paper, while alphabetical citations refer to new references.

---

### Decision · Program_Chairs · 2021-09-27

**Decision:**

Accept (Poster)

**Comment:**

Visualization with UMAP has become standard practice in multiple fields in recent years, originally presented as being derived from topological construction as an alternative to the popular tSNE visualization. However, over the past few years several studies have investigated its relation with tSNE (and other visualizations) and raised questions regarding the matching (or rather discrepancy) between its theoretical derivation and the implementation used in practice. This work provides an interesting in depth analysis on this topic, constructively providing a closed-form formulation of the loss function that is implemented by the UMAP algorithm (which is different from the loss it was purportedly supposed to optimize). In doing so, it provides concise insights into understanding of previous observations, and more generally the behavior of UMAP.

After discussion, the reviewers unanimously agree the paper should be accepted. Most concerns that have been raised by initial reviews have been addressed in the responses (authors: please follow up on these in revision of the manuscript itself). Therefore, I recommend accepting the paper and I am positively certain it will be of great interest to a significant portion of the NeurIPS community.